# Beyond Dirichlet-based Models: When Bayesian Neural Networks Meet Evidential Deep Learning

**Hanjing Wang**[1]                    **Qiang Ji**[1]

[1]ECSE Dept., Rensselaer Polytechnic Institute, Troy, New York, USA

## Abstract

Bayesian neural networks (BNNs) excel in uncertainty quantification (UQ) by estimating the posterior distribution of model parameters, yet face challenges due to the high computational demands of Bayesian inference. Evidential deep learning methods address this by treating target distribution parameters as random variables with a learnable conjugate distribution, enabling efficient UQ. However, there's debate over whether these methods can accurately estimate epistemic uncertainty due to their single-network, sampling-free nature. In this paper, we combine the strengths of both approaches by distilling BNN knowledge into a Dirichlet-based model, endowing it with a Bayesian perspective and theoretical guarantees. Additionally, we introduce two enhancements to further improve the integration of Bayesian UQ with Dirichlet-based models. To relax the heavy computational load with BNNs, we introduce a self-regularized training strategy using Laplacian approximation (LA) for self-distillation. To alleviate the conjugate prior assumption, we employ an expressive normalizing flow for refining the model in a post-processing manner, where a few training iterations can enhance model performance. The experimental results have demonstrated the effectiveness of our proposed methods in both UQ accuracy and robustness.

## 1 INTRODUCTION

Deep neural networks (DNNs) have shown remarkable performance in a wide range of machine learning applications, particularly with the advent of big data. However, conventional DNNs may exhibit overconfidence and fail to accurately model the uncertainty associated with predictions. This shortcoming could lead to catastrophic outcomes in safety-critical real-world applications. Hence, it is imperative to develop reliable systems employing probabilistic models that incorporate UQ, thus enabling well-informed and confident decision-making.

There are two types of uncertainties: epistemic and aleatoric uncertainty. Epistemic uncertainty refers to the prediction uncertainty arising from limited knowledge in the modeling process, while aleatoric uncertainty stems from the inherent noise present in the data. To accurately quantify both types of uncertainties, Bayesian neural networks (BNNs) can be employed, which treat NN parameters as random variables and compute the posterior distribution of these parameters. Common approaches for UQ include Markov Chain Monte Carlo (MCMC) methods [Tierney, 1994, Welling and Teh, 2011, Chen et al., 2014b, Zhang et al., 2019] and variational inference (VI) techniques [Louizos and Welling, 2017, Maddox et al., 2019, Franchi et al., 2020]. Ensemble-based methods [Lakshminarayanan et al., 2017, Valdenegro-Toro, 2019, Wen et al., 2020] also provide powerful alternatives for achieving accurate UQ. Despite their advantages, traditional BNNs require substantial and diverse parameter samples for Bayesian inference, leading to inefficiency in UQ. This inefficiency stems from the necessity to carry out multiple forward propagations of the NN using distinct parameters sampled from the posterior. To address this issue, researchers have introduced single-network deterministic methods that facilitate the computation of uncertainty via a single forward pass of the network.

Evidential deep learning methods [Ulmer, 2021] assume that the target variable in classification problems follows a categorical distribution, with parameters treated as random variables governed by a conjugate Dirichlet distribution, thus facilitating epistemic uncertainty quantification. However, current approaches necessitate additional knowledge to accurately learn the parameters of the conjugate distribution, such as out-of-distribution (OOD) data [Malinin and Gales, 2018, 2019], ensemble models [Malinin et al., 2019], and density models [Charpentier et al., 2020]. Even with

this added knowledge, point-estimated evidential networks can struggle to quantify epistemic uncertainty. Malinin and Gales [2018] and Ulmer [2021] argued that mutual information based uncertainty primarily reflects distributional uncertainty instead of epistemic uncertainty. Charpentier et al. [2020] and Sensoy et al. [2018] proposed using the sharpness of the Dirichlet distribution to measure epistemic uncertainty. However, these methods diverge from entropy-based UQ and often lack theoretical justification for their uncertainty measures. These challenges drive our exploration of evidential networks from a Bayesian perspective, ensuring the capture of epistemic uncertainty with theoretical guarantees. Furthermore, we introduce two enhancements to complement our core idea, offering alternative opportunities for achieving a better accuracy-efficiency trade-off. Our contributions are summarized below:

- We propose to incorporate BNN knowledge into an evidential network, endowing it with a Bayesian perspective and theoretical guarantees. The uncertainty can be estimated through a single pass of the NN.

- A self-distilled training strategy using Laplacian approximation is introduced, eliminating the need for additional knowledge of BNNs and reducing the computational burden.

- We suggest using flexible normalizing flows in a post-processing way to relax conjugate prior assumptions, improving UQ with few training iterations.

## 2 RELATED WORKS

**Bayesian neural networks** BNNs treat NN parameters as random variables, seeking to determine their posterior distribution. Since Bayesian inference entails marginalizing over the posterior, finding a closed-form solution for Bayesian inference becomes challenging. Therefore, different methods for approximately generating parameter samples from the posterior are proposed. The MCMC methods [Tierney, 1994, Welling and Teh, 2011, Chen et al., 2014b, Zhang et al., 2019] construct Markov chains to align their equilibrium distribution with the posterior distribution, which in turn provides a basis for sampling. VI methods [Louizos and Welling, 2017, Maddox et al., 2019, Franchi et al., 2020], on the other hand, learn a simpler distribution that approximates the posterior from which samples can be drawn during inference. As a special variational method, the Laplace approximation (LA) method [MacKay, 1992] approximates the posterior distribution using a Gaussian distribution by performing Taylor expansion around its mode, which can be applied to any pre-trained probabilistic neural networks (PNNs)[1] for UQ. To improve the efficiency of LA, sub-network LA [Daxberger et al., 2021b] and last-layer LA

[Kristiadi et al., 2020] are proposed which construct the posterior distribution for subsets of NN parameters. Meanwhile, the deep ensemble method [Lakshminarayanan et al., 2017] and its variants [Wen et al., 2020, Wenzel et al., 2020, Han et al., 2020, Li et al., 2022] generate multiple samples of NN parameters by training with varying random initializations, serving as strong baselines.

**Evidential deep learning and deterministic UQ** Evidential deep learning techniques treat the parameters of the target distribution as random variables adhering to a conjugate distribution. To learn the conjugate distribution, additional knowledge often comes from OOD training samples, where various approaches [Malinin and Gales, 2018, 2019, Nandy et al., 2020, Shen et al., 2020, Chen et al., 2018, Sensoy et al., 2020] enforce a flat Dirichlet distribution over the OOD data. Other regularizations for accurate conjugate distribution learning involve the use of ensemble models for knowledge distillation [Malinin et al., 2019], the accumulation of evidence in the subjective logic framework [Sensoy et al., 2018], and the density models of latent variables [Charpentier et al., 2020]. These methods have distinct loss functions, such as Kullback–Leibler (KL) divergence loss [Malinin and Gales, 2018], L2 norm loss [Sensoy et al., 2018], and ELBO loss [Chen et al., 2018]. Additionally, some approaches [Amini et al., 2020, Malinin et al., 2020, Charpentier et al., 2021] extend the Dirichlet-based methods to regression problems. For more detailed discussions, please refer to Appendix B.1 and the survey paper [Ulmer, 2021]. Recently, some deterministic UQ methods have been proposed that do not require a Dirichlet framework. These methods include those that construct distance-aware hidden representations for uncertainty quantification through gradient penalty [Van Amersfoort et al., 2020] or spectral normalization [Lakshminarayanan et al., 2020, van Amersfoort et al., 2021], and those that learn more informative hidden representations through contrastive learning [Wu and Goodman, 2020] or reconstruction regularization [Postels et al., 2020]. However, these techniques often have poorly calibrated uncertainty [Postels et al., 2021] and struggle to distinguish between aleatoric and epistemic uncertainty.

**Bayesian knowledge distillation** Hinton et al. [2015] pioneered Bayesian knowledge distillation by transferring knowledge from complex ensemble models to a single deterministic counterpart. Subsequent works by Korattikara Balan et al. [2015] and Englesson and Azizpour [2019] extended this concept, distilling BNN knowledge into deterministic networks. However, these methods often omit epistemic uncertainty information. Recently, Lindqvist et al. [2020] focused on distilling the ensemble distribution in the latent space. Unlike evidential deep learning, their UQ approach requires sampling, whereas our method calculates epistemic uncertainty within a single forward pass.

---

[1]We define a PNN as a neural network that outputs the probability distribution of the target variable.

## 3 PRELIMINARIES

**General notations and assumptions.** We denote the input as $x$, the target variable as $y$, and the training data as $\mathcal{D} = \{x_n, y_n\}_{n=1}^N$. In this work, we focus on classification problems, where $y$ is a random variable following a categorical distribution. E represents expectation, H represents entropy, and I represents mutual information. $\beta$ is a hyperparameter for determining the prior distribution of a random variable. $C$ is the number of classes.

**Bayesian neural networks** Bayesian neural networks assume the NN parameters $\psi$ are random variables, with a prior $p(\psi|\beta)$ and a likelihood $p(\mathcal{D}|\psi)$. We can apply the Bayes' rule to compute the posterior of $\psi$, i.e., $p(\psi|\mathcal{D}, \beta)$, as shown in Eq. (1).

$$p(\psi \mid \mathcal{D}, \beta) = \frac{p(\mathcal{D} \mid \psi)p(\psi \mid \beta)}{p(\mathcal{D})} \tag{1}$$

For classification problems, we assume the target $y$ follows a categorical distribution with the parameter $\lambda(x, \psi)$. The posterior of $\lambda$ can be computed as follows:

$$p(\lambda \mid x, \mathcal{D}, \beta) = \int p(\lambda \mid x, \psi)p(\psi \mid \mathcal{D}, \beta)d\psi \tag{2}$$

where $p(\lambda \mid x, \psi) = \delta(\lambda(x, \psi))$ and $\delta(\cdot)$ is a Dirac probability density function. Given a new testing sample $x^*$, Bayesian inference marginalizes out $\psi$ over its posterior $p(\psi|\mathcal{D}, \beta)$ to obtain $p(y|x^*, \mathcal{D}, \beta)$, as shown in Eq. (3).

$$p(y \mid x^*, \mathcal{D}, \beta) = \int p(y \mid x^*, \psi)p(\psi \mid \mathcal{D}, \beta)d\psi \tag{3}$$

Finally, the entropy-based UQ of a BNN is shown in the following:

$$\underbrace{\mathrm{H}[p(y|x, \mathcal{D}, \beta)]}_{\text{Total } U_t} = \underbrace{\mathrm{I}[y; \psi|x, \mathcal{D}, \beta]}_{\text{Epistemic } U_e} + \underbrace{\mathrm{E}_{p(\psi|\mathcal{D}, \beta)}\{\mathrm{H}[p(y|x, \psi)]\}}_{\text{Aleatoric } U_a}. \tag{4}$$

**Laplacian approximation** Given a MAP-trained PNN with parameters $\psi$ such that $\psi_{map} = \arg\max_\psi \log p(\psi|\mathcal{D}, \beta)$, LA approximates $p(\psi|\mathcal{D}, \beta)$ by a Gaussian distribution $\mathcal{N}(\psi_{map}, \Sigma)$ with mean $\psi_{map}$ and covariance matrix $\Sigma$ as shown in Eq. (5):

$$p(\psi|\mathcal{D}, \beta) \approx \mathcal{N}(\psi_{map}, \Sigma); \ \ \Sigma = -(G)^{-1} \tag{5}$$

where $G = \nabla_\psi^2 \log p(\psi|\mathcal{D}, \beta)|_{\psi=\psi_{map}}$. To achieve high efficiency, we use last-layer LA [Kristiadi et al., 2020] in this paper. More discussions can be found in Appendix A.

**Dirichlet-based model** For classification problems, the target $y$ follows a categorical distribution with parameter $\lambda$, i.e., $y \sim p(y \mid \lambda) = Cat(\lambda)$. The parameter of the categorical distribution $\lambda$ is also treated as a random variable, following a Dirichlet distribution, i.e., $\lambda \sim p(\lambda \mid \alpha(x, \theta)) =$

$Dir(\alpha(x, \theta))$, where $\alpha(x, \theta) = [\alpha_1, \alpha_2, \cdots, \alpha_C]^T$ is the output of a deterministic NN parameterized by $\theta$. The probability $p(y|x, \theta)$ is expressed as:

$$p(y \mid x, \theta) = p(y \mid \alpha(x, \theta)) = \int p(y \mid \lambda)p(\lambda \mid \alpha)d\lambda$$
$$= Cat\left(\left\{\frac{\alpha_k}{\alpha_0}\right\}_{k=1}^C\right) \tag{6}$$

where $C$ is the number of classes and $\alpha_0 = \sum_{k=1}^C \alpha_k$. In the hierarchical model $\theta \to \alpha \to \lambda \to y$, only $\theta$ is learnable, with other parameters explicitly calculated. Similar to standard neural network training, adding Dirichlet $\lambda$ doesn't complicate the neural network and the training process due to closed-form integration. To clarify, training directly with the likelihood in Eq. (6) resembles standard training, aiming to align $p(y|\alpha(x, \theta))$ with ground truth labels. However, it can only solve for $\alpha$ up to a scale factor $\alpha_0$. Determining $\alpha_0$ controls the sharpness of the Dirichlet distribution, which is important for epistemic UQ.

## 4 PROPOSED METHOD

In this section, we first introduce Dir-BNN, a Dirichlet-based framework that performs accurate and efficient UQ by integrating the BNN knowledge. Then, we propose a self-regularized training strategy by LA to train Dir-BNN more efficiently. Lastly, we employ a normalizing flow to refine the Dirichlet distribution in a post-processing manner.

### 4.1 DIR-BNN TRAINING: A BAYESIAN PERSPECTIVE

To learn the Dir-BNN parameters $\theta$ under the Dirichlet-based model illustrated in Sec. 3, the Maximum-A-Posteriori (MAP) loss is represented in Eq. (7), with $p(y|x, \theta)$ given in Eq. (6) and a pre-defined prior $p(\theta)$:

$$\mathcal{L}_{map}(\theta) = -\sum_{(x,y)\in\mathcal{D}} \log p(y|x, \theta) - \log p(\theta) \tag{7}$$

Nonetheless, based on Eq. (6), the MAP estimation can only determine $\alpha$ up to the scale factor $\alpha_0 = \sum_k \alpha_k$. We suggest utilizing both the ground truth training data and the BNN knowledge as regularization to train the Dir-BNN jointly. The total loss function can be expressed as follows:

$$\mathcal{L}(\theta) = \mathcal{L}_{map}(\theta) + \rho\mathcal{L}_{reg}(\theta). \tag{8}$$

In this case, $\mathcal{L}_{reg}(\theta)$ represents the regularization loss function that incorporates supervision from the BNN model. $\rho$ is the coefficient of the regularization, serving as a hyperparameter. To extract knowledge from a BNN, the Dir-BNN learns Dirichlet distribution $p(\lambda \mid \alpha(x, \theta)) = Dir(\alpha(x, \theta))$ to approximate the posterior distribution $p(\lambda \mid x, \mathcal{D}, \beta)$

derived from the BNN model in Eq. (2). As a result, we can employ KL-divergence between $p(\lambda \mid x, \mathcal{D}, \beta)$ and $p(\lambda \mid \alpha(x, \theta))$ as the regularization term:

$$
\begin{aligned}
\mathcal{L}_{reg}(\theta) =&\, \mathrm{KL}\left(p(\lambda \mid x, \mathcal{D}, \beta)\|p(\lambda \mid \alpha(x, \theta))\right) \\
\propto&\, -\sum_{k=1}^{C} \log\left(\Gamma(\alpha_k)\right) + \log\Gamma\left(\sum_{k=1}^{C}\alpha_k\right) \\
&\, - \mathrm{E}_{p(\psi|\mathcal{D},\beta)}\left[\sum_{k=1}^{C}(\alpha_k - 1)\log\lambda_k(x, \psi)\right].
\end{aligned}
$$
(9)

The derivation of Eq. (9) is available in Appendix B.2. Training Dir-BNN with just the KL divergence loss is also feasible with a perfect BNN. However, with a suboptimal BNN, adding NLL loss may improve Dir-BNN's predictions and ensure stable training. As noted in Sec.3.1 of [Malinin et al., 2019], Dirichlet-based methods may encounter optimization challenges. Optimizing Eq. (9) at the start of training is difficult because the Bayesian posterior for the output space is often "sharp" at one corner of the simplex. Initially, when the NN is randomly initialized, the Dirichlet distribution's mode is near the center, making the KL optimization challenging due to limited common support between the two distributions. Adding the NLL loss can guide the optimization in the right direction at the beginning of the training. After training the Dir-BNN, the uncertainties can be calculated within a single forward pass of the NN without the need for sampling. We estimate the epistemic and aleatoric uncertainties using mutual information and expected entropy, respectively, as demonstrated in Eq. (10):

$$
\underbrace{\mathrm{H}[p(y|x, \theta)]}_{\text{Total } U_t} = \underbrace{\mathrm{I}[y; \lambda|\alpha]}_{\text{Epistemic } U_e} + \underbrace{\mathrm{E}_{p(\lambda|\alpha)}\left[\mathrm{H}[p(y|\lambda)]\right]}_{\text{Aleatoric } U_a}.
$$
(10)

The closed-form expressions for the uncertainties mentioned above can be found in Appendix B.3. Incorporating the Bayesian regularization term leads to several significant improvements. It endows the Dirichlet models with a Bayesian perspective, which emulates the process of quantifying epistemic uncertainty through variations in model parameters. Conventional evidential UQ methods, utilizing point-estimated models, struggle to effectively quantify epistemic uncertainty. As indicated by Malinin and Gales [2018], the uncertainty estimated by $\mathrm{I}[y; \lambda|\alpha]$ can only capture distributional uncertainty. In contrast, our proposed approach can be considered a variational method for performing Bayesian inference in the output space. The following propositions offer theoretical guarantees for our methods in quantifying epistemic uncertainty, with proofs provided in Appendix B.4.

**Proposition 4.1.1** (Performing UQ in the output space). *For a BNN $f$ with parameter $\psi \sim p(\psi|\mathcal{D}, \beta)$, which outputs the softmax probability, i.e., $\lambda = f(x, \psi)$, we have $\mathrm{I}[y; \psi|x, \mathcal{D}] = \mathrm{I}[y; \lambda|x, \mathcal{D}]$.*

**Proposition 4.1.2** (Transformation of the variational gap). *Given a BNN as defined in Proposition 4.1.1, we assume there is a variational approximation $q_\theta(\psi)$ for approximating $p(\psi|\mathcal{D}, \beta)$. Correspondingly, the posterior distribution $p(\lambda|\mathcal{D}, x, \beta)$ is approximated by $q_\theta(\lambda|x)$, which fulfills:*

$$
q_\theta(\lambda|x) = \int \delta(\lambda = f(x, \psi))q_\theta(\psi)d\psi \qquad (11)
$$

*where $\delta(\cdot)$ is the Dirac probability density function. The KL divergence between $p(\lambda|\mathcal{D}, x, \beta)$ and $q_\theta(\lambda|x)$ is upper bounded by KL divergence between $p(\psi|\mathcal{D}, \beta)$ and $q_\theta(\psi)$:*

$$
KL(p(\lambda|\mathcal{D}, x, \beta)||q_\theta(\lambda|x)) \le c + KL(p(\psi|\mathcal{D}, \beta)||q_\theta(\psi)).
$$
(12)

*$c$ is a constant with respect to $\theta$ as shown in Eq. (13). The equality is satisfied when $f$ is invertible.*

$$
\begin{aligned}
c = -H_\lambda + H_\psi =&\, \int p(\lambda|\mathcal{D}, x, \beta)\log p(\lambda|\mathcal{D}, x, \beta)d\lambda \\
&\, - \int p(\psi|\mathcal{D}, \beta)\log p(\psi|\mathcal{D}, \beta)d\psi
\end{aligned}
$$
(13)

Since the prediction uncertainty is determined in the output space, Proposition 4.1.1 establishes that calculating epistemic uncertainty in the output space using the posterior distribution of $\lambda$ is both adequate and necessary. Since $p(\lambda|x, \mathcal{D}, \beta)$ is often intractable, Proposition 4.1.2 shows that it is more advantageous to learn a variational distribution to approximate $p(\lambda|x, \mathcal{D}, \beta)$ than to conduct VI in the parameter space. It emphasizes that the output-space variational gap $(\mathrm{KL}(p(\lambda|\mathcal{D}, x, \beta)||q_\theta(\lambda|x)))$ has an upper bound determined by the parameter-space variational gap $(\mathrm{KL}(p(\psi|\mathcal{D}, \beta)||q_\theta(\psi)))$. As a result, minimizing the distance between $p(\psi|\mathcal{D}, \beta)$ and $q_\theta(\psi)$ yields a sub-optimal approximation for $p(\lambda|\mathcal{D}, x, \beta)$. In this paper, we consider the Dir-BNN as an approximate inference method bridging Bayesian deep learning and evidential deep learning. Fundamentally, we learn a Dirichlet distribution in the output space to approximate $p(\lambda|\mathcal{D}, x, \beta)$. Thus, our method offers theoretical guarantees for accurate epistemic UQ. Proposition 4.1.3 illustrates that the error in UQ is constrained by the output-space variational gap. After adding the regularization term $\mathrm{KL}(p(\lambda|\mathcal{D}, x, \beta)||p(\lambda|x, \theta))$, we can theoretically show that the estimated uncertainties from Dir-BNN serve as approximations to their exact measurements.

**Proposition 4.1.3** (Uncertainty error bound). *Consider $U_t, U_a, U_e$ as the total, aleatoric, and epistemic uncertainties determined by the Dir-BNN, as depicted in Eq. (10). Similarly, $U_t^*, U_a^*, U_e^*$ denote their counterparts derived from the BNN as presented in Eq. (4). Under certain conditions (will be shown in Eq. (55) of Appendix B.4.3), the following error bounds demonstrate the distance between the uncertainties estimated by a Dir-BNN and their exact measurements from the BNN:*

$$|U_t - U_t^*| \leq -\sqrt{2\log 2 \cdot d} \log \frac{\sqrt{2\log 2 \cdot d}}{C}$$

$$|U_a - U_a^*| \leq \log C \sqrt{2\log 2 \cdot d}$$

$$|U_e - U_e^*| \leq \sqrt{2\log 2 \cdot d} \left(2\log C - \log\sqrt{2\log 2 \cdot d}\right) \tag{14}$$

*where $d = KL(p(\lambda|\mathcal{D}, x, \beta)||p(\lambda|x, \theta))$.*

We further present two analyses for the uncertainty error bounds: (1) under the assumptions illustrated in Eq. (55) of Appendix B.4.3, all bounds decrease as $d$ decreases; (2) all bounds approach 0 as $d \to 0$. It is evident that the bound for $U_a$ satisfies the above conditions. Since the bound for $U_e$ is the sum of the bounds for $U_t$ and $U_a$ (i.e., we use $|U_e - U_e^*| \leq |U_t - U_t^*| + |U_a - U_a^*|$ for Proposition 4.1.3), we only need to demonstrate that the bound for $U_t$ meets these conditions. Based on Lemma B.2 in Appendix B.4.3, the function $-x \log \frac{x}{C}$ (with $x = \sqrt{2\log 2d}$ for the bound of $U_t$) is monotonically non-decreasing if $x \in [0, \frac{C}{e}]$. Therefore, under the assumptions for Proposition 4.1.3, the bound of $U_t$ is non-decreasing. As $d$ approaches 0, the bound tends to 0, according to L'Hôpital's Rule:

$$\lim_{x \to 0} -x \log \frac{x}{C} = \lim_{x \to 0} \frac{\log \frac{C}{x}}{\frac{1}{x}} = \lim_{x \to 0} \frac{\frac{d}{dx}\log\frac{C}{x}}{\frac{d}{dx}\frac{1}{x}}$$
$$= \lim_{x \to 0} \frac{-1/x}{-1/x^2} = \lim_{x \to 0} x = 0 \tag{15}$$

It's also worth noting that the assumptions for Proposition 4.1.3 provide an upper bound for $d$, which may generally hold as $p(\lambda|x, \theta)$ is trained effectively to approximate $p(\lambda|x, \mathcal{D}, \beta)$. $d$ often does not "explode" since the distributions are defined on the probability simplex in the output space, which is much smaller than the parameter space.

Although the proposed method requires a BNN, it is only necessary during training. While training may take longer, the resulting BNN-augmented evidential network facilitates fast and accurate UQ with theoretical guarantees. Given the intractability of exact BNNs, we can utilize approximate BNNs like VI or ensemble methods for supervision, leading to the creation of the Dir-ESB, an ensemble-augmented Dirichlet model. For example, we can approximate BNNs using $S$ samples $\psi^1, \ldots, \psi^S$ through the ensemble method. To train Dir-ESB, we replace the expectation term in Eq. (9) with a sample average:

$$E_{p(\psi|\mathcal{D},\beta)}\left[\sum_{k=1}^{C}(\alpha_k - 1)\log\lambda_k(x, \psi)\right]$$
$$\approx \frac{1}{S}\sum_{s=1}^{S}\sum_{k=1}^{C}(\alpha_k - 1)\log\lambda_k(x, \psi^s) \tag{16}$$

## 4.2 SELF-DISTILLATION TRAINING STRATEGY

Acquiring the exact posterior distribution in a BNN can be challenging. To mitigate the training complexity associated with the BNN, we use LA as a self-regularized training approach for the Dirichlet-based model. When training the Dirichlet-based model without regularization, the posterior distribution of the NN parameters can be approximated using LA, based on existing values of the parameters. Subsequently, adding the LA regularization loss allows for further model refinement. This uncertainty regularization stems directly from the single-network Dirichlet model itself. Details can be found in Algorithm 1. It is worth noting that in step 3 of Algorithm 1, we apply LA on $\theta$ given the likelihood $p(y|x, \theta) = p(y|\alpha(x, \theta))$ in Eq. (6) and a prior $p(\theta)$, aligning closely with standard LA.

---

**Algorithm 1** Dir-LA: Pseudocode for Dir-BNN with Self-regularization

---

1: Input: training data $\mathcal{D}$, Dir-LA model $\theta$;
2: Use the MAP loss in Eq. (7) to train a Dirichlet-based model;
3: Perform the LA, i.e., $p(\psi|\mathcal{D}, \beta) \approx \mathcal{N}(\theta_{map}, \Sigma)$ following Eq. (5);
4: Given the approximated posterior distribution $\mathcal{N}(\theta_{map}, \Sigma)$, refine the current model using the total loss shown in Eq. (8) where the regularization loss is shown in Eq. (19);

---

Moreover, LA allows us to obtain a closed-form expression of Eq. (9) without the need of sampling during training. Denote the output logits of the BNN as $f(x, \psi)$ and $\lambda(x, \psi) = softmax(f(x, \psi))$. We perform the first-order Taylor expansion for $f(x, \psi)$ with respect to $\psi$ at $\psi = \psi_{map}$:

$$f(x, \psi) \approx f(x, \psi_{map}) + J_f^T(\psi - \psi_{map}) \tag{17}$$

where $J_f = \nabla_\psi f(x, \psi)|_{\psi=\psi_{map}}$ is the Jacobian matrix of $f(x, \psi)$ with respect to $\psi$ at $\psi = \psi_{map}$. Hence, given $\psi \sim \mathcal{N}(\psi_{map}, \Sigma)$ from LA, $f(x, \psi)$ also follows a Gaussian distribution, i.e., $f(x, \psi) \sim \mathcal{N}(f_{map}, \Sigma_f)$, where $f_{map} = f(x, \psi_{map})$ and $\Sigma_f = J_f \Sigma J_f^T$.

Denote $l(f(x, \psi)) = \sum_{k=1}^{C}(\alpha_k - 1)\log\lambda_k(x, \psi)$. We take the second-order Taylor expansion of $l(f(x, \psi))$ with respect to $f(x, \psi)$ at $f = f_{map}$ as shown in Eq. (18).

$$l(f) \approx l(f_{map}) + J_l^T(f - f_{map}) + \frac{1}{2}(f - f_{map})^T H_l(f - f_{map}) \tag{18}$$

Then, the regularization loss in Eq. (9) can be solved in a closed-form manner by substituting Eq. (18) into Eq. (9) as shown in Eq. (19), where $tr$ represents the trace of a matrix. The detailed expressions of $l(f_{map})$, $J_l$, $H_l$ and the

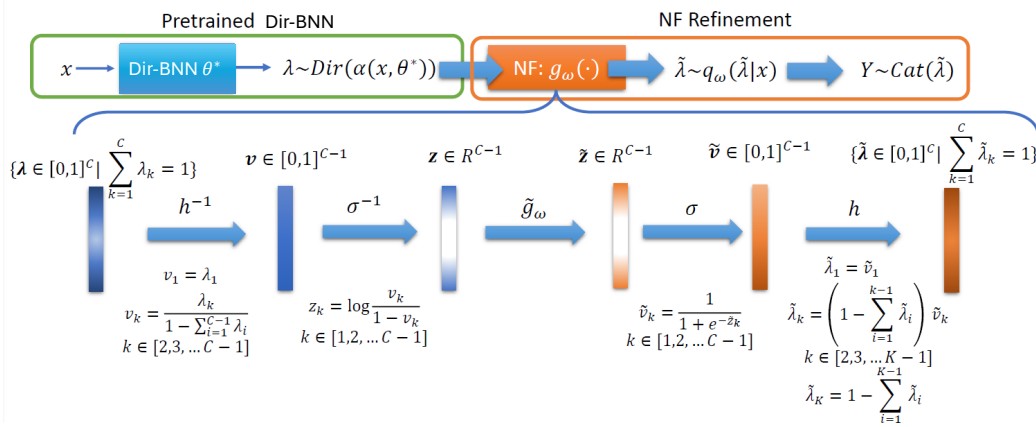

Figure 1: The pipeline of Dir-BNN refinement using NF.

derivation of Eq. (19) can be found in Appendix C.

$$\mathcal{L}_{reg}(\theta) = -\sum_{k=1}^{C} \log(\Gamma(\alpha_k)) + \log \Gamma(\sum_{k=1}^{C} \alpha_k)$$
$$- l(f_{map}) - \frac{1}{2} tr(H_l \Sigma_f) \quad (19)$$

### 4.3 DIR-BNN REFINEMENT USING NORMALIZING FLOW

Normalizing flows (NFs) have gained prominence as a potent method for augmenting the expressiveness and adaptability of variational inference in Bayesian deep learning. Essentially, NFs are parametric generative models capable of generating tractable density functions. Starting from a simple distribution (e.g., standard normal), NFs transform it into a more intricate distribution through a series of invertible and differentiable transformations. Gordon-Rodriguez et al. [2020] highlighted several drawbacks of the Dirichlet distribution: it complicates optimization for compositional data models, its log-likelihood becomes undefined when observations contain zeros, and the maximum likelihood estimation of its mean parameter is biased. Hence, we employ NFs to achieve a more expressive distribution than the Dirichlet distribution for superior posterior approximation. As Proposition 4.1.3 demonstrates, more expressive distributions can result in better posterior approximation and improved UQ.

Instead of learning a normalizing flow for the distribution of $\lambda$ at the outset of Dir-BNN training, we incorporate NFs in a post-processing fashion to refine the acquired Dirichlet distribution. Upon completing the Dir-BNN training as per Sec. 4.1, we proceed to learn an NF, commencing from the derived Dirichlet distribution, for a more accurate posterior approximation. The post-processing NF does not affect the efficient UQ by Dir-BNN but offers alternative opportunities for balancing UQ accuracy and efficiency.

The entire process and the architecture of the NF are depicted in Figure 1. It is important to note that traditional

NFs are designed for continuous variables, necessitating the development of a specialized NF for transforming simplex-valued variables. To achieve this, we first employ the invertible function $h^{-1}$ to convert the simplex-valued $\lambda$ into $v \in [0,1]^{C-1}$. Subsequently, the inverse of the sigmoid function $\sigma^{-1}$ is utilized to transform $v$ into a real-valued variable $z$. The traditional NF for continuous variables, $\tilde{g}_\omega$, is then applied. Lastly, the functions $h$ and $\sigma$ are used in the corresponding order to transform the real-valued $\tilde{z}$ into our desired $\tilde{\lambda}$. The design of $h, h^{-1}, \sigma, \sigma^{-1}$ is detailed in Figure 1 and Appendix D.1. Given the invertible function $g_\omega = h \circ \sigma \circ \tilde{g}_\omega \circ \sigma^{-1} \circ h^{-1}$, $q_\omega(\tilde{\lambda})$ has a tractable density function based on the change of variable property:

$$q_\omega(\tilde{\lambda}) = Dir(g_\omega^{-1}(\lambda); \alpha(x, \theta^*)) \left| \det J_{g_\omega^{-1}}(\tilde{\lambda}) \right| \quad (20)$$

where $\det J_{g_\omega^{-1}}(\tilde{\lambda})$ is the determinant for the Jacobian matrix of $g_\omega^{-1}$ with respect to $\tilde{\lambda}$, which will be detailed in Appendix D.2. Finally, $\omega$ is learned by minimizing the distance between $p(\tilde{\lambda}|\mathcal{D}, x, \beta)$ and $q_\omega(\tilde{\lambda})$ as shown in Eq. (21). $p(\tilde{\lambda}|\mathcal{D}, x, \beta)$ is equivalent to $p(\lambda|\mathcal{D}, x, \beta)$ as both $\tilde{\lambda}$ and $\lambda$ denote the simplex-valued probability.

$$\omega^* = \arg \min_\omega \mathrm{KL}(p(\tilde{\lambda}|\mathcal{D}, x, \beta) || q_\omega(\tilde{\lambda}))$$
$$= \arg \min_\omega - \mathrm{E}_{\tilde{\lambda} \sim \tilde{\lambda}|\mathcal{D}, x, \beta}[\log q_\omega(\tilde{\lambda})] \quad (21)$$

By directly optimizing Eq. (21) and starting from the Dirichlet posterior, the NF distribution is guaranteed to be closer to the true Bayesian posterior.

## 5 EXPERIMENTS

**Dataset** In this paper, we evaluate using benchmark image classification datasets: MNIST [Deng, 2012], Fashion-MNIST (FMNIST) [Xiao et al., 2017], CIFAR-10 (C10) [Krizhevsky et al., 2014], and CIFAR-100 (C100) [Krizhevsky et al., 2009].

**Implementation details** We employ standard CNNs for MNIST and FMNIST and Resnet18 for C10 and C100. The

Table 1: OOD detection results for AUROC (%) ↑ and AUPR (%) ↑ with epistemic uncertainty. "*" represents our method.

| Method | MNIST → Omniglot | | MNIST → KMNIST | | FMNIST→ EMNIST | | FMNIST→ MNIST | | Avg |
|---|---|---|---|---|---|---|---|---|---|
| | AUROC | AUPR | AUROC | AUPR | AUROC | AUPR | AUROC | AUPR | |
| PriorNet | 94.5 ± .65 | 95.0 ± .43 | 98.5 ± .05 | 98.0 ± .11 | 73.5 ± .90 | **96.9** ± .12 | 83.4 ± 1.2 | 96.1 ± .84 | 91.99 |
| PosNet | 96.4 ± .36 | 96.1 ± .38 | 97.8 ± .35 | 97.2 ± .33 | 83.8 ± 2.6 | 92.7 ± 1.1 | 86.3 ± 3.3 | 88.8 ± 2.3 | 92.39 |
| EvNet | 71.8 ± 5.3 | 81.3 ± 3.7 | 18.6 ± 3.6 | 42.4 ± 2.5 | 84.1 ± 3.8 | 91.8 ± 2.3 | 83.8 ± 4.0 | 84.8 ± 3.6 | 69.83 |
| DUQ | 97.3 ± .33 | **97.8** ± .17 | **98.7** ± .43 | **98.9** ± .33 | 79.8 ± 1.1 | 82.6 ± .78 | 92.4 ± 1.3 | 92.4 ± 1.0 | 92.49 |
| DUE | 94.3 ± .18 | 94.1 ± .54 | 93.5 ± .43 | 93.4 ± .10 | 92.4 ± .96 | 93.1 ± 1.2 | 92.1 ± .45 | 92.5 ± .37 | 93.18 |
| LA | 94.5 ± .25 | 93.3 ± .42 | 94.9 ± .26 | 93.0 ± .31 | 88.9 ± .73 | 90.9 ± .79 | **96.5** ± .38 | **96.9** ± .91 | 93.61 |
| Ensemble | 97.5 ± .20 | 97.3 ± .25 | 98.5 ± .05 | 98.4 ± .05 | **93.0** ± .45 | 93.7 ± .65 | 90.8 ± .14 | 95.4 ± .99 | 95.58 |
| Dir-LA* | **97.9** ± .16 | 97.7 ± .17 | **98.7** ± .21 | 98.5 ± .25 | 91.8 ± .89 | 95.2 ± .23 | 94.4 ± .14 | 93.9 ± .73 | 96.01 |
| Dir-ESB* | 97.7 ± .07 | 97.6 ± .05 | 98.5 ± .10 | 98.3 ± .10 | 92.0 ± .21 | 93.3 ± 1.1 | 92.0 ± .88 | 92.3 ± 1.3 | 95.21 |
| Dir-ESB-NF* | **97.9** ± .20 | 97.7 ± .21 | **98.7** ± .20 | 98.5 ± .27 | 92.8 ± .62 | 95.8 ± .20 | 95.5 ± .53 | 94.9 ± .51 | **96.48** |

| Method | C10 → SVHN | | C10 → LSUN | | C100→ SVHN | | C100→ LSUN | | Avg |
|---|---|---|---|---|---|---|---|---|---|
| | AUROC | AUPR | AUROC | AUPR | AUROC | AUPR | AUROC | AUPR | |
| PriorNet | 64.5 ± 9.7 | **91.1** ± 3.0 | 73.9 ± 2.4 | 68.2 ± 4.3 | 44.8 ± .42 | **85.5** ± 1.2 | 51.7 ± 3.5 | 52.9 ± 2.9 | 66.58 |
| PosNet | 84.6 ± 2.1 | 80.1 ± 1.3 | 87.2 ± .71 | 86.6 ± 1.1 | 63.8 ± 8.0 | 57.2 ± 6.9 | 45.2 ± 1.4 | 45.9 ± 2.3 | 68.83 |
| EvNet | 59.7 ± 2.0 | 67.3 ± 2.4 | 61.0 ± 1.9 | 68.9 ± 2.2 | 51.6 ± .30 | 48.6 ± 1.2 | 50.3 ± 1.2 | 48.1 ± .28 | 56.94 |
| DUQ | 86.2 ± 1.1 | 80.1 ± .98 | 87.9 ± .77 | 85.6 ± 1.2 | 40.9 ± 1.3 | 42.0 ± 1.5 | 55.6 ± 1.2 | 52.9 ± 1.2 | 66.40 |
| DUE | 82.2 ± 1.5 | 76.0 ± 1.7 | 79.3 ± .55 | 78.9 ± .57 | 64.1 ± 1.9 | 63.9 ± 1.3 | 64.8 ± .49 | 60.9 ± .83 | 71.26 |
| LA | 88.3 ± .22 | 85.7 ± .47 | 87.1 ± .34 | 84.6 ± 1.0 | 80.1 ± .12 | 73.6 ± .39 | 76.5 ± .40 | 71.9 ± .48 | 80.98 |
| Ensemble | 89.5 ± .15 | 84.2 ± .38 | 86.9 ± .34 | 83.1 ± 1.0 | **81.5** ± .87 | 76.2 ± 1.1 | 78.4 ± .74 | 73.2 ± 1.2 | 81.63 |
| Dir-LA* | 89.6 ± .74 | 82.9 ± 1.9 | 89.1 ± .97 | 82.1 ± 2.1 | 80.6 ± 3.1 | 75.5 ± 3.0 | 76.8 ± 3.6 | **74.2** ± 3.1 | 81.35 |
| Dir-ESB* | 87.6 ± 1.0 | 82.6 ± .60 | **90.4** ± .60 | 85.7 ± 1.7 | 80.9 ± 4.4 | 75.3 ± 4.0 | 79.5 ± .61 | 71.7 ± .83 | 81.71 |
| Dir-ESB-NF* | **90.6** ± .74 | 89.0 ± 1.9 | 90.1 ± .97 | **88.3** ± 2.1 | 81.4 ± 3.3 | 76.8 ± 2.2 | **80.6** ± 1.9 | 73.8 ± 1.6 | **83.82** |

three methods tested include Dir-ESB, which uses ensemble models (as a BNN) for regularization as per Sec. 4.1; Dir-LA, which implements the self-regularization for efficient training as per Sec. 4.2; Dir-ESB-NF, a post-processing refinement of Dir-ESB utilizing NF as per Sec. 4.3. We chose to refine Dir-ESB using NF instead of Dir-LA. This decision helps maintain the essential information from LA. It also prevents the NF-generated distribution from aligning too closely with LA, especially since LA serves as a strong assumption for posterior approximation. Note that Dir-ESB-NF requires only a shallow NF with few training iterations. Appendix E contains all experiment configurations and implementation details.

**Baselines** We compare our proposed methods against various baselines including Dirichlet-based models (PriorNet [Malinin and Gales, 2018], PosNet [Charpentier et al., 2020], EvNet [Sensoy et al., 2018]), Deterministic UQ methods (DUQ [Van Amersfoort et al., 2020], DUE [van Amersfoort et al., 2021]), the deep ensemble method (Ensemble) [Lakshminarayanan et al., 2017], and the last-layer LA (LA) [Kristiadi et al., 2020]. Ensemble and LA denote the BNNs' performances guiding the supervision of Dir-ESB and Dir-LA, respectively.

**Evaluation tasks** In this section, we assess the proposed method through OOD detection and uncertainty calibration in the face of distributional shifts.

## 5.1 OUT-OF-DISTRIBUTION DETECTION

Out-of-distribution detection, one of the primary applications for UQ [Malinin and Gales, 2018, Malinin et al., 2019], aims to detect anomalous data inconsistent with the training data distribution, using a measure of uncertainty. Since epistemic uncertainty is negatively correlated with data density, it will be high when predicting anomalous data.

**Experiment settings** For the MNIST dataset, we obtain OOD samples from Omniglot [Lake et al., 2015], KMNIST [Clanuwat et al., 2018]. In the case of the FMNIST dataset, OOD samples are drawn from both the MNIST and EMNIST [Cohen et al., 2017] datasets. The OOD datasets used for comparison with the CIFAR-10 and CIFAR-100 datasets are SVHN Netzer et al. [2011] and LSUN Yu et al. [2015]. For OOD detection, we use two evaluation metrics: the area under the receiver operating characteristic curve (AUROC) and the area under the precision-recall curve (AUPR).

**Experiment results and analysis** The OOD detection outcomes are presented in Table 1. Our methods consistently outperform existing Dirichlet-based and deterministic UQ methods, with an average 10% - 15% enhancement on the C10 and C100 datasets. Dir-ESB, despite distilling knowledge from the Ensemble method, achieves results on par with the Ensemble. This might be attributed to the framework of fitting a Dirichlet distribution, guided by the ensemble model modes, which capture not only mode information but also the entire landscape of the posterior approximation, enabling sampling parameters from the neighborhood of the modes. Dir-LA also performs competitively compared to LA, possibly due to the Dirichlet-based hierarchical structure and the specifically designed loss function in Eq. (19). Given that Dir-LA achieves competitive results and boasts superior training efficiency compared to Dir-ESB, we recommend employing Dir-LA as a sufficient solution for OOD detection tasks. Dir-ESB-NF, built using an NF to refine Dir-ESB, shows noticeable improvement compared to Dir-ESB, particularly on the FMNIST and C10 datasets. For more OOD detection results, please refer to Sec. 5.2 for detecting

Table 2: Uncertainty calibration performance for ACC (%) ↑, ALC ↑, and ROC (%) ↑. ROC represents AUROC.

| Method | Rotate 20 | | | Rotate 60 | | | Rotate 100 | | | Avg | | |
|---|---|---|---|---|---|---|---|---|---|---|---|---|
| | ACC | ALC | ROC | ACC | ALC | ROC | ACC | ALC | ROC | ACC | ALC | ROC |
| PriorNet | 95.10 | 0.91 | 68.20 | 31.90 | 0.36 | 94.63 | 16.35 | 0.21 | 96.10 | 47.78 | 0.49 | 86.31 |
| PosNet | 95.19 | 0.91 | 63.51 | 32.75 | **0.50** | 92.15 | 14.69 | **0.29** | 92.83 | 47.54 | **0.57** | 82.83 |
| EvNet | 95.65 | 0.02 | 44.80 | 34.87 | 0.07 | 36.92 | 17.16 | 0.02 | 40.91 | 49.22 | 0.04 | 40.88 |
| DUQ | 90.91 | 0.90 | 66.30 | 27.56 | 0.33 | 90.92 | 15.62 | 0.09 | 86.71 | 44.70 | 0.47 | 81.31 |
| DUE | 94.15 | **0.92** | 69.10 | 31.03 | 0.37 | 96.53 | 12,67 | 0.13 | 96.96 | 45.95 | 0.47 | 87.53 |
| LA | 93.53 | 0.87 | 69.29 | 35.43 | 0.38 | 93.22 | 18.12 | 0.13 | 94.64 | 49.02 | 0.46 | 85.71 |
| Ensemble | **96.18** | **0.92** | 68.52 | 34.39 | 0.29 | 96.10 | 15.65 | 0.15 | 96.50 | 48.74 | 0.45 | 87.04 |
| Dir-LA* | **96.18** | 0.90 | 70.00 | **35.51** | 0.34 | 96.84 | **18.68** | 0.17 | 97.08 | **50.12** | 0.47 | 87.97 |
| Dir-ESB* | 95.70 | 0.92 | 73.76 | 33.64 | 0.32 | 97.46 | 17.38 | 0.13 | 97.77 | 48.91 | 0.46 | 89.66 |
| Dir-ESB-NF* | 95.94 | **0.92** | 73.78 | 33.76 | 0.38 | **97.58** | 17.53 | 0.16 | **97.83** | 49.08 | 0.49 | **89.73** |

| Method | Noise 0.05 | | | Noise 0.1 | | | Noise 0.15 | | | Avg | | |
|---|---|---|---|---|---|---|---|---|---|---|---|---|
| | ACC | ALC | ROC | ACC | ALC | ROC | ACC | ALC | ROC | ACC | ALC | ROC |
| PriorNet | 62.21 | 0.66 | 53.91 | 54.09 | 0.59 | 65.07 | 41.10 | 0.49 | 76.71 | 52.47 | 0.58 | 65.23 |
| PosNet | 69.63 | 0.63 | 69.32 | 32.46 | 0.39 | 89.99 | 17.44 | 0.20 | **93.41** | 39.84 | 0.41 | 84.24 |
| EvNet | 73.95 | 0.11 | 46.24 | 40.17 | 0.14 | 52.17 | 24.64 | 0.17 | 62.25 | 46.25 | 0.14 | 53.55 |
| DUQ | 81.81 | 0.82 | 58.98 | 44.43 | 0.46 | 76.22 | 22.67 | 0.29 | 85.15 | 49.63 | 0.52 | 73.45 |
| DUE | 81.15 | 0.64 | 53.13 | 56.88 | 0.48 | 61.23 | 35.93 | 0.27 | 66.06 | 57.99 | 0.46 | 60.14 |
| LA | 70.66 | 0.72 | **73.55** | 33.78 | 0.42 | **91.13** | 18.07 | 0.24 | 92.16 | 40.84 | 0.46 | **85.88** |
| Ensemble | 77.92 | 0.75 | 70.94 | 32.07 | 0.29 | 83.32 | 14.69 | 0.28 | 85.93 | 41.56 | 0.44 | 80.06 |
| Dir-LA* | 71.26 | 0.69 | 72.03 | 34.50 | 0.46 | 86.94 | 22.20 | 0.29 | 91.88 | 42.65 | 0.48 | 83.62 |
| Dir-ESB* | 71.32 | 0.73 | 71.97 | 34.34 | 0.46 | 86.77 | 21.85 | 0.30 | 92.60 | 42.50 | 0.50 | 83.78 |
| Dir-ESB-NF* | **82.40** | **0.84** | 60.49 | **70.49** | **0.75** | 72.06 | **58.35** | **0.64** | 80.75 | **70.41** | **0.74** | 71.10 |

varying levels of shifted data.

## 5.2 UNCERTAINTY CALIBRATION ANALYSIS

In this section, we will showcase the efficacy of our methods in handling image classification tasks with synthetic distributional shifts, with a focus on uncertainty calibration. As demonstrated by Postels et al. [2021], numerous deterministic uncertainty quantification methods lack proper calibration. Consequently, a comprehensive analysis of uncertainty calibration for our proposed techniques is vital.

**Experiment settings** We develop shifted datasets, namely rotated MNIST and noisy CIFAR-10. Specifically, we modify the MNIST test data by rotating it from $0°$ to $180°$ in $20°$ increments and create the noisy CIFAR-10 dataset by introducing Gaussian noise with mean 0 and variance ranging from 0 to 0.25 in 0.05 steps. Various metrics are employed, including accuracy (ACC), the relative area under the lift curve (ALC) as introduced by Postels et al. [2021], and AUROC for OOD detection tasks MNIST → rotated MNIST and C10 → noisy C10. Note that ALC is a calibration metric that analyzes how well the uncertainty score aligns with the prediction error. We excluded the estimation of Expected Calibration Error (ECE) and Negative Log-Likelihood (NLL) from our evaluation for all baselines because these metrics do not apply to DUQ and DUE, which do not yield softmax probabilities directly. In Appendix F, the within-dataset performance, additional metrics such as ECE and NLL for other baselines, and more experiment settings are provided.

**Experiment results and analysis** As displayed in Table 2, Dir-ESB and Dir-LA continue to outperform Dirichlet-based and deterministic UQ methods on the MNIST dataset. While PriorNet, DUQ, and DUE demonstrate strong ACC/ALC performance on the C10 dataset, Dir-ESB and Dir-LA maintain competitive performance across all metrics when compared to SOTAs. Owing to the Dirichlet-based architecture and the unique training strategy, Dir-ESB and Dir-LA on average perform competitively compared to Ensemble and LA. Comparing Dir-LA to Dir-ESB highlights Dir-LA's comparable performance and superior efficiency in incorporating BNN knowledge. For the MNIST dataset, Dir-ESB-NF enhances Dir-ESB across all metrics, while improving ACC and ALC for the C10 dataset. This refinement involves only a shallow NF and a few training iterations.

## 5.3 ABLATION STUDIES

**Runtime/Complexity analysis** Dir-ESB requires pre-trained ensemble models, and once these are in place, its training time aligns with that of a standard NN. Compared to training a standard NN, Dir-LA requires additional time for last-layer LA, $O(C^3 + P^3)$, where $C$ is the class count and $P$ is the last-layer parameter count. However, this doesn't notably increase training time since it's a one-time process, marginal over the entire training duration. For inference, both Dir-ESB and Dir-LA can achieve high efficiency, estimating uncertainty in a single forward pass. Our Dir-ESB-NF only utilizes a shallow NF. By starting with a pre-trained Dir-ESB, the convergence only requires a few training iterations, not substantially increasing training or inference time. A comprehensive complexity analysis and empirical runtime are detailed in Appendix G.1.

**Refining output space vs. parameter space** In this section, we compare our method with Kristiadi et al. [2022]. While both studies use a normalizing flow for improved posterior approximation, refining the Dirichlet distribution in the output space offers distinct theoretical merits, as high-

lighted by Proposition 4.1.2. This proposition reveals that a good parameter-space approximation doesn't guarantee a comparable output-space approximation. Given that prediction uncertainty is calculated in the output space, an accurate approximation there is crucial. Experimentally, we adapted Kristiadi et al. [2022] 's method to refine the last-layer LA in weight space for the MNIST and C10 datasets. The OOD detection performance (AUROC/AUPR) is shown in Table 3. Our output-space NF refinement outperforms the refinement conducted in the parameter space.

Table 3: OOD detection performance.

| Method | MNIST → Omniglot | | C10 → SVHN | |
|---|---|---|---|---|
| | AUROC | AUPR | AUROC | AUPR |
| Dir-LA | 97.9 | 97.7 | 89.6 | 82.9 |
| Kristiadi et al. | 97.6 | 96.6 | 89.1 | 82.7 |

**Comparison with standard knowledge distillation**  A standard deterministic NN can only capture the aleatoric uncertainty, while the Dirichlet-based framework can capture both types of uncertainty. Nevertheless, we follow Korattikara et al. [2015] to perform distillation between $p(y|x, \mathcal{D})$ and $p(y|x, \theta)$ where $p(y|x, \mathcal{D})$ is approximated by ensemble models. The uncertainty is quantified by the entropy, which primarily denotes the aleatoric uncertainty. Our OOD detection results (AUROC/AUPR) for MNIST → Omniglot and C10 → SVHN outperform the standard distillation (97.7/96.6 vs 97.7/97.5 and 87.6/82.6 vs 86.2/80.9).

**Hyperparameter sensitivity**  A crucial hyperparameter for our method is the regularization loss coefficient, denoted as $\rho$ in Eq. (8). This hyperparameter balances supervised training and knowledge distillation from the BNN. $\rho$ should not be excessively small or large. A too-small $\rho$ will fail to effectively distill knowledge from the BNN, while an overly large $\rho$ might negatively impact classification performance. To scrutinize the effect of $\rho$, we conduct OOD detection on the MNIST and C10 datasets using Dir-LA. For selected values of $\rho$ from 25 to 65, we empirically illustrate in Table 4 that our methods are relatively insensitive to the values of $\rho$ within a certain range. Empirically, adjusting the balance between MAP and regularization losses may enhance model performance.

Table 4: OOD detection results (AUROC (%)/AUPR (%)) for varying $\rho$ between 25 to 65.

| Method | MNIST → Omniglot | | C10 → SVHN | |
|---|---|---|---|---|
| | AUROC | AUPR | AUROC | AUPR |
| Dir-LA $\rho = 25$ | 97.6 | 97.5 | 89.5 | 82.7 |
| DIr-LA $\rho = 30$ | 98.7 | 98.5 | 89.5 | 82.8 |
| Dir-LA $\rho = 40$ | 97.8 | 97.6 | 89.6 | 82.9 |
| Dir-LA $\rho = 50$ | 97.9 | 97.7 | 89.6 | 82.9 |
| Dir-LA $\rho = 65$ | 98.1 | 98.1 | 89.5 | 82.7 |

**NF architecture and training**  In this study, we only require a shallow flow to refine the Dirichlet distribution, serving two purposes: it enhances efficiency, and a shallow NF is sufficient for refining the model. We do not seek perfect alignment with the approximate inference methods

used for distillation; thus, a shallow flow adequately extracts critical information from the approximate posterior without overfitting. We empirically evaluate the OOD performance of Dir-ESB-NF by varying the number of flows, with results shown in Table 5. Increasing the NF's complexity does not yield notable improvements. Given the shallow NF and the initially trained Dir-BNN, only about 200 iterations are necessary. More discussions are shown in Appendix G.2.

Table 5: OOD detection results for AUROC (%) ↑ and AUPR (%) ↑ for Dir-ESB-NF with varying number of flows.

| Method | MNIST → Omniglot | | C10 → SVHN | |
|---|---|---|---|---|
| | AUROC | AUPR | AUROC | AUPR |
| 10-layer flow | 97.9 | 97.7 | 89.6 | 82.9 |
| 15-layer flow | 97.9 | 97.7 | 89.6 | 82.9 |
| 20-layer flow | 97.9 | 97.7 | 89.6 | 82.9 |

**Other BNNs for knowledge distillation**  Dir-BNN is compatible with many BNN approximations. We chose the ensemble and LA methods for their effectiveness in UQ. To further demonstrate Dir-BNN's effectiveness, we explore its combination with SGLD [Welling and Teh, 2011] and SGHMC [Chen et al., 2014a]. MCMC methods are powerful techniques to approximate the posterior distribution if the burn-in period is long enough and sufficient samples are generated. For implementing SGLD and SGHMC, we followed the experimental settings outlined in Appendix E.1 to ensure consistency in model architecture and certain training protocols. For the burn-in period, we used 50 epochs. After the burn-in period, we selected 50 samples from each method as supervision for the Dirichlet-based model. The OOD detection performance (AUROC/AUPR) on MNIST and FMNIST datasets is shown in Table 6. It is shown that Dir-SGLD and Dir-SGHMC also perform competitively.

Table 6: OOD detection results for AUROC (%) ↑ and AUPR (%) ↑ for our methods with various BNN supervision.

| Method | MNIST → Omniglot | | MNIST → KMNIST | |
|---|---|---|---|---|
| | AUROC | AUPR | AUROC | AUPR |
| Dir-ESB | 97.7 | 97.6 | 98.5 | 98.3 |
| Dir-SGLD | 97.9 | 97.3 | 98.4 | 98.3 |
| Dir-SGHMC | 98.0 | 98.0 | 97.4 | 97.3 |

| Method | FMNIST → EMNIST | | FMNIST → MNIST | |
|---|---|---|---|---|
| | AUROC | AUPR | AUROC | AUPR |
| Dir-ESB | 92.0 | 93.3 | 92.0 | 92.3 |
| Dir-SGLD | 94.1 | 97.3 | 94.6 | 95.7 |
| Dir-SGHMC | 90.4 | 95.7 | 91.5 | 93.5 |

# 6  CONCLUSION

In this paper, we introduce a Dirichlet-based framework for accurate and efficient UQ by incorporating BNN knowledge. Then, a self-regularized training strategy using LA is proposed to relax the requirement of BNNs for knowledge distillation, which empirically shows strong performance. Finally, we also present an opportunity to boost model per-

formance via post-processing NF refinement. Various experiments on OOD detection and uncertainty calibration analysis have demonstrated the effectiveness and superiority of our proposed methods.

**Acknowledgement.** This work is supported in part by the National Science Foundation award IIS 2236026.

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

# A  LAPLACIAN APPROXIMATION

The LA shown in Eq. (5) approximates the posterior distribution $p(\psi|\mathcal{D}, \beta)$ by $\mathcal{N}(\psi_{map}, \Sigma)$ where $\Sigma = -(H)^{-1}$ and $H = \nabla_\psi^2 \log p(\psi|\mathcal{D}, \beta)|_{\psi=\psi_{map}}$. Efficiently and accurately calculating the Hessian matrix $H$ is the key of LA. Given a standard Gaussian distribution prior $p(\psi) = \mathcal{N}(0, \beta^2 I)$ where $\beta$ is the hyperparameter, we can obtain that

$$
\begin{aligned}
\nabla_\psi^2 \log p(\psi|\mathcal{D}, \beta) =& \nabla_\psi^2 \log p(\mathcal{D}|\psi) + \nabla_\psi^2 \log p(\psi|\beta) \\
=& \sum_{(x,y)\in\mathcal{D}} \nabla_\psi^2 \log p(y|x,\psi) + \frac{1}{\beta^2} I
\end{aligned}
\tag{22}
$$

where $I$ is the identity matrix. Basically, computing the second-order derivatives for highly nonlinear neural networks is hard and we leverage the Generalized Gauss-Newton Matrix (GGN) [Schraudolph, 2002] to approximate $\nabla_\psi^2 \log p(y|x,\psi)$. Denote the neural network output as $f(x, \psi)$ in general.

$$
\begin{aligned}
\nabla_\psi^2 \log p(y|x,\psi) =& \nabla_\psi^2 \log p(y|f(x,\psi)) \\
\approx& J(x)\nabla_f^2 p(y|f(x,\psi))J(x)^T
\end{aligned}
\tag{23}
$$

where $J(x) = \nabla_\psi f(x, \psi)$ is the Jacobian matrix. However, the large matrix multiplication in Eq. (23) may also lead to problems. We use the last-layer Laplacian approximation proposed by Kristiadi et al. [2020]. To reduce computational complexity, different Hessian matrix factorization methods are also proposed such as Kronecker-factored approximation curvature (KFAC) [Ritter et al., 2018] and low-rank KFAC [Lee et al., 2020]. We use the full Hessian matrix for MNIST, FMNIST, and C10 datasets. To improve efficiency, we use a diagonal covariance matrix for the C100 dataset. To avoid tuning the hyperparameter $\beta$, we utilize the marginal likelihood maximization method proposed by Ritter et al. [2018] to do a one-parameter optimization for $\beta$. The loss function is the posterior predictive approximated by LA.

$$
\beta^* = \arg\max_\beta \sum_{(x,y)\in\mathcal{D}} \log p(y|x, \mathcal{D})
\tag{24}
$$

where the posterior $p(y|x, \mathcal{D})$ can be expressed as:

$$
\begin{aligned}
p(y|x, \mathcal{D}) =& \int p(y|x,\psi)p(\psi|\mathcal{D},\beta)d\psi \\
\approx& \int softmax(f(x,\psi))\mathcal{N}(\psi; \psi_{map}, \Sigma)d\psi
\end{aligned}
\tag{25}
$$

$softmax(f) = \frac{exp(f)}{\sum_j exp(f_j)}$ is the softmax function and Eq. (25) can be solved either by MC sample average or by probit approximation [Daxberger et al., 2021a].

# B  DIRICHLET NETWORK TRAINING WITH BNN DISTILLATION

In this section, we will delve deeper into our distinctive contribution of distilling BNN knowledge for Dirichlet-based model training, detailing the aspects of uncertainty and providing comprehensive proofs for all the theorems presented.

## B.1  MORE DISCUSSIONS ON OUR CONTRIBUTIONS

As outlined in [Ulmer, 2021], Dirichlet-based evidential deep learning techniques can be classified into two types: prior Dirichlet networks and posterior Dirichlet networks. Prior Dirichlet networks directly learn the parameters of the Dirichlet distribution, treating it as a conjugate prior to the desired categorical distribution. The form of the Dirichlet distribution $p(\lambda|x,\theta) = p(\lambda|\alpha(x,\theta))$ is as follows:

$$
\begin{aligned}
Dir(\lambda; \alpha(x,\theta)) =& \frac{1}{B(\alpha(x,\theta))} \prod_{k=1}^C \lambda_k^{\alpha_k-1} \\
B(\alpha(x,\theta)) =& \frac{\prod_{k=1}^C \Gamma(\alpha_k)}{\Gamma(\sum_{i=1}^C \alpha_i)}
\end{aligned}
\tag{26}
$$

Conversely, posterior Dirichlet networks learn $p(\lambda|x, \mathcal{D}, \theta)$, which is depicted in Eq. (27):

$$
\begin{aligned}
p(\lambda|x, \mathcal{D}, \theta) \propto& p(\mathcal{D}|x, \lambda, \theta)p(\lambda|x, \theta) \\
=& p(\{y_i\}_{i=1}^N|x, \lambda, \theta)p(\lambda|x, \theta) \\
=& \prod_{i=1}^N \prod_{k=1}^C \lambda_k^{I_{y_i=k}} \cdot \frac{1}{B(\alpha(x,\theta))} \prod_{k=1}^C \lambda_k^{\alpha_k-1} \\
=& \frac{1}{B(\alpha(x,\theta))} \prod_{k=1}^C \lambda_k^{\alpha_k+N_k-1} \\
\propto& Dir(\lambda; \{\alpha_k + N_k - 1\}_{k=1}^C) \\
=& Dir(\lambda; \alpha^{prior} + \alpha^{data})
\end{aligned}
\tag{27}
$$

where $\alpha^{data}$ is determined by $\{N_k\}_{k=1}^C$ and $N_k$ represents the counts for $y_i = k$ in $\{y_i\}_{i=1}^N$. Given an input $x$, there is usually just one label $y$ available. Therefore, $\{y_i\}_{i=1}^N$ is computed using pseudo counts approximated by a latent generative model, such as a normalizing flow [Charpentier et al., 2020]. However, modeling the density of the lower-dimensional latent space necessitates additional effort, and the density in the latent space might not adequately represent the density in the input space. As we do not use an estimate of $\alpha$ from the training data density, our methods fall under the category of prior Dirichlet networks.

Prior Dirichlet networks primarily vary in the regularization terms within their loss function. As seen in related studies, some techniques [Malinin and Gales, 2018, 2019, Nandy

et al., 2020, Shen et al., 2020, Chen et al., 2018, Sensoy et al., 2020] aim to flatten the Dirichlet distribution on OOD samples. However, it is unrealistic to assume that OOD samples are always available, since enumerating all possible OOD data configurations during training is impossible. Consequently, the learned model might overfit to one type of OOD data seen during training, failing to generalize to other OOD data types. Some methods [Sensoy et al., 2018] attempt to regularize training by diminishing the density with respect to incorrect classes in the Dirichlet distribution. They often diverge from others in terms of loss functions and employ L2 loss instead of negative log-likelihood loss. The method that is most similar to ours is ensemble distribution distillation [Malinin et al., 2019], which learns the Dirichlet distribution by distilling knowledge from Ensemble models. A self-distilled strategy [Fathullah and Gales, 2022] has been proposed to learn the teacher model and the Dirichlet network simultaneously, sharing a feature extractor and efficiently generating multiple NN outputs using the Gaussian dropout method.

Our proposed method seeks to employ Bayesian neural networks for knowledge distillation, resulting in efficient and accurate uncertainty quantification. It can be seen as a strict extension of [Malinin et al., 2019]. The primary advantage of distilling knowledge from a BNN is that it offers a theoretical assurance for the Dirichlet network to estimate epistemic uncertainty. The posterior predictive distribution for a BNN is displayed as follows:

$$
\text{BNN:} \quad p(y|x, \mathcal{D}, \beta) = \int \underbrace{p(y|x, \theta)}_{aleatoric} \underbrace{p(\theta|\mathcal{D}, \beta)}_{epistemic} \, d\theta
$$
$$
= \int \underbrace{p(y|\lambda)}_{aleatoric} \underbrace{p(\lambda|x, \mathcal{D}, \beta)}_{epistemic} \, d\lambda \tag{28}
$$

where $p(\lambda|x, \mathcal{D}, \beta) = \int p(\lambda|x, \theta)p(\theta|\mathcal{D}, \beta)d\lambda$ as shown in Eq. (2). Since we are interested in the prediction uncertainty, our proposed method directly approximates $p(\lambda|x, \mathcal{D}, \beta)$ under the Dirichlet framework, as shown below:

$$
\text{Dirichlet network:} \quad p(y|x, \mathcal{D}, \beta)
$$
$$
= \int p(y|\lambda) \underbrace{p(\lambda|x, \mathcal{D}, \beta)}_{\approx Dir(\alpha(x, \theta^*))} \, d\lambda \tag{29}
$$

Our approach carries out variational inference under the Dirichlet assumption by incorporating the regularization term $KL(p(\lambda|x, \mathcal{D}, \beta)||Dir(\alpha(x, \theta)))$. This procedure specifically encapsulates the epistemic uncertainty instigated by the weight posterior. The associated theorems are presented in Sec. 4.1. Moreover, as depicted by Malinin and Gales [2018], a universal Dirichlet-based network solely encapsulates distribution uncertainty:

General Dirichlet Network:

$$
p(y|x, \mathcal{D}, \beta) = \int \underbrace{p(y|\lambda)}_{aleatoric} \underbrace{p(\lambda|x, \theta)}_{distributional} \underbrace{p(\theta|\mathcal{D}, \beta)}_{epistemic} \, d\lambda d\theta
$$
$$
= \int \underbrace{p(y|\lambda)}_{aleatoric} \underbrace{p(\lambda|x, \theta^*)}_{distributional} \, d\lambda
$$
$$
\tag{30}
$$

where $p(\theta|\mathcal{D}, \beta) \approx \delta(\theta = \theta^*)$. The absence of a direct correlation between $p(\lambda|x, \theta^*)$ and $p(\lambda|x, \mathcal{D}, \beta)$ hampers its capability to compute epistemic uncertainty. it is important to note that distributional uncertainty may impart some epistemic uncertainty information. Unless explicitly designed using our methods, the main distinction lies in the distributional uncertainty's inability to identify the uncertainty captured by the weight posterior. Although some techniques claim to capture epistemic uncertainty via the sharpness factor of the Dirichlet distribution, we contend that such uncertainties are predominantly distributional in nature, as they are merely properties of $p(\lambda|x, \theta^*)$ and do not consider the weight posterior. Based on Proposition 4.1.1, the epistemic uncertainty can be computed in the following:

$$
\underbrace{\text{H}\left\{\text{E}_{p(\lambda|x, \mathcal{D}, \beta)}[p(y|\lambda)]\right\}}_{\text{Total uncertainty}}
$$
$$
= \underbrace{\text{I}[y; \lambda|x, \mathcal{D}, \beta]}_{\text{Epistemic uncertainty}} + \underbrace{\text{E}_{p(\lambda|x, \mathcal{D}, \beta)}\{\text{H}[p(y|\lambda)]\}}_{\text{Aleatoric uncertainty}}. \tag{31}
$$

In the absence of an explicit approximation $p(\lambda|x, \mathcal{D}, \beta) \approx p(\lambda|\mathcal{D}, \theta^*) = Dir(\alpha(x, \theta^*))$, substituting $p(\lambda|x, \mathcal{D}, \beta)$ with $p(\lambda|\mathcal{D}, \theta^*)$ is less effective in capturing the epistemic uncertainty.

In our study, we employ LA and ensemble models as BNN approximations. It is important to note that while the Dirichlet network's performance is influenced by BNN regularization, it is not necessarily limited by the approximate BNN models' performance. As seen in the Sec. 5, the Dirichlet-based model can effectively extract crucial information from the approximate posterior distribution while disregarding some potentially misleading aspects. This could be due to the Dirichlet framework and our uniquely crafted loss function.

## B.2 DERIVATION OF EQ. (9)

The derivation of Eq. (9) is shown below:

$$
\begin{aligned}
\mathcal{L}_{reg}(\theta) &= \text{KL}\left(p(\lambda \mid x, \mathcal{D}, \beta) \| p(\lambda \mid \alpha(x, \theta))\right) \\
&\propto -\int p(\lambda \mid x, \mathcal{D}, \beta) \log p(\lambda \mid \alpha(x, \theta)) d\lambda \\
&= -\int\int p(\lambda \mid x, \psi)\, p(\psi \mid \mathcal{D}, \beta)[\log p(\lambda \mid \alpha(x, \theta))] d\lambda d\psi \\
&= -\int p(\psi \mid \mathcal{D}, \beta)[\log p(\lambda(x, \psi) \mid \alpha(x, \theta))] d\psi \\
&= -\sum_{k=1}^{K} \log(\Gamma(\alpha_k)) + \log \Gamma(\sum_{k=1}^{K} \alpha_k) \\
&\quad - E_{p(\psi|\mathcal{D},\beta)}\left[\sum_{k=1}^{K} (\alpha_k - 1) \log \lambda_k(x, \psi)\right]
\end{aligned}
\tag{32}
$$

## B.3 UNCERTAINTY QUANTIFICATION OF DIRICHLET NETWORK

As demonstrated in Sec. 4.1, we evaluate the epistemic and aleatoric uncertainties using mutual information and expected entropy, respectively:

$$
\underbrace{\text{H}[p(y|x, \theta)]}_{\text{Total uncertainty}} = \underbrace{\text{I}[y; \lambda|\alpha]}_{\text{Epistemic uncertainty}} + \underbrace{E_{p(\lambda|\alpha)}\left[\text{H}[p(y|\lambda)]\right]}_{\text{Aleatoric uncertainty}}.
\tag{33}
$$

The closed-form expressions of Eq. (33) are shown in Eq. (34).

$$
\text{H}[p(y|x, \theta)] = -\sum_{k=1}^{C} \frac{\alpha_k}{\alpha_0} \log \frac{\alpha_k}{\alpha_0}
$$

$$
\text{I}[y; \lambda|\alpha] = -\sum_{k=1}^{C} \frac{\alpha_k}{\alpha_0}\left(\ln \frac{\alpha_k}{\alpha_0} - \Psi(\alpha_k + 1) + \Psi(\alpha_0 + 1)\right)
\tag{34}
$$

where $\alpha_0 = \sum_{k=1}^{C} \alpha_k$ and $\Psi(\cdot)$ is the dgamma function. The derivations of Eq. (34) can be found in Appendix C of [Malinin and Gales, 2018].

## B.4 PROOF OF THE THEOREMS

### B.4.1 Proposition 4.1.1:

*For a BNN $f$ with parameter $\psi \sim p(\psi|\mathcal{D}, \beta)$ that outputs the softmax probability $\lambda = f(x, \psi)$, we have $\text{I}[y; \psi|x, \mathcal{D}] = \text{I}[y; \lambda|x, \mathcal{D}]$.*

*Proof.* As shown in Eq. (4):

$$
\underbrace{\text{H}\left\{E_{p(\psi|\mathcal{D},\beta)}[p(y|x, \psi)]\right\}}_{\text{Total uncertainty}}
$$
$$
= \underbrace{\text{I}[y; \psi|x, \mathcal{D}, \beta]}_{\text{Epistemic uncertainty}} + \underbrace{E_{p(\psi|\mathcal{D},\beta)}\{\text{H}[p(y|x, \psi)]\}}_{\text{Aleatoric uncertainty}}.
\tag{35}
$$

By introducing $\lambda$ in the BNN framework:

$$
\begin{aligned}
&\text{H}\left\{E_{p(\psi|\mathcal{D},\beta)}[p(y|x, \psi)]\right\} \\
&= \text{H}\left\{E_{p(\psi|\mathcal{D},\beta)}[E_{p(\lambda|x,\psi)}\, p(y|\lambda)]\right\} \\
&= \text{H}\left\{E_{p(\lambda|x,\mathcal{D},\beta)}[p(y|\lambda)]\right\} \\
&E_{p(\psi|\mathcal{D},\beta)}\{\text{H}[p(y|x, \psi)]\} \\
&= E_{p(\psi|\mathcal{D},\beta)}\{\text{H}[p(y|\lambda(x, \psi))]\} \\
&= E_{p(\lambda|\mathcal{D},\beta)}\{\text{H}[p(y|\lambda)]\}.
\end{aligned}
\tag{36}
$$

Given that

$$
\underbrace{\text{H}\left\{E_{p(\lambda|x,\mathcal{D},\beta)}[p(y|\lambda)]\right\}}_{\text{Total uncertainty}}
$$
$$
= \underbrace{\text{I}[y; \lambda|x, \mathcal{D}, \beta]}_{\text{Epistemic uncertainty}} + \underbrace{E_{p(\lambda|x,\mathcal{D},\beta)}\{\text{H}[p(y|\lambda)]\}}_{\text{Aleatoric uncertainty}}.
\tag{37}
$$

and compare Eq. (36) with Eq. (37), we can finally obtain:

$$
\text{I}[y; \psi|x, \mathcal{D}] = \text{I}[y; \lambda|x, \mathcal{D}].
\tag{38}
$$

$\square$

### B.4.2 Proposition 4.1.2:

*Given a BNN defined in Proposition 4.1.1, we assume there is a variational approximation $q_\theta(\psi)$ for approximating $p(\psi|\mathcal{D}, \beta)$. Correspondingly, the posterior distribution $p(\lambda|\mathcal{D}, x, \beta)$ is approximated by $q_\theta(\lambda|x)$, which fulfills:*

$$
q_\theta(\lambda|x) = \int \delta(\lambda = f(x, \psi)) q_\theta(\psi) d\psi
\tag{39}
$$

*where $\delta(\cdot)$ is the Dirac probability density function. The KL divergence between $p(\lambda|\mathcal{D}, x, \beta)$ and $q_\theta(\lambda|x)$ is upper bounded by KL divergence between $p(\psi|\mathcal{D}, \beta)$ and $q_\theta(\psi)$:*

$$
KL(p(\lambda|\mathcal{D}, x, \beta)\|q_\theta(\lambda|x)) \le c + KL(p(\psi|\mathcal{D}, \beta)\|q_\theta(\psi)).
\tag{40}
$$

*$c$ is a constant with respect to $\theta$ as shown in Eq. (41). The equality is satisfied when $f$ is invertible.*

$$
\begin{aligned}
c &= -H_\lambda + H_\psi = \int p(\lambda|\mathcal{D}, x, \beta) \log p(\lambda|\mathcal{D}, x, \beta) d\lambda \\
&\quad - \int p(\psi|\mathcal{D}, \beta) \log p(\psi|\mathcal{D}, \beta) d\psi
\end{aligned}
\tag{41}
$$

*Proof.* For a BNN defined in Proposition 4.1.1, the simplex-valued probability $\lambda$ is the output of the BNN by $\lambda = f(x, \psi)$. Given the posterior distribution of $\psi$, i.e., $p(\psi|\mathcal{D}, \beta)$, or its variational approximation $q_\theta(\psi)$,

$$p(\lambda|x, \mathcal{D}, \beta) = \int \delta(\lambda = f(x, \psi = \psi')) p(\psi = \psi'|\mathcal{D}, \beta) d\psi'$$

$$q_\theta(\lambda|x) = \int \delta(\lambda = f(x, \psi = \psi'')) q_\theta(\psi = \psi'') d\psi''$$

(42)

where $\psi'$ and $\psi''$ are specific values of random variable $\psi$. Then, we can obtain

$$\begin{aligned}
&\mathrm{KL}(p(\lambda|\mathcal{D}, x, \beta)||q_\theta(\lambda|x)) \\
&= -H_\lambda - \mathrm{E}_{\lambda' \sim p(\lambda|x, \mathcal{D}, \beta)}[\log q_\theta(\lambda = \lambda'|x)] \\
&= -H_\lambda - \mathrm{E}_{\psi' \sim p(\psi|\mathcal{D}, \beta)} \\
&\quad \left\{ \mathrm{E}_{\lambda' \sim \delta(\lambda = f(x, \psi = \psi'))} \left[ \log q_\theta(\lambda = \lambda'|x) \right] \right\} \\
&= -H_\lambda - \mathrm{E}_{\psi' \sim p(\psi|\mathcal{D}, \beta)} \left[ \log q_\theta(\lambda = f(x, \psi = \psi')|x) \right] \\
&= -H_\lambda - \mathrm{E}_{\psi' \sim p(\psi|\mathcal{D}, \beta)} \\
&\quad \left[ \log \int \delta(f(x, \psi = \psi') = f(x, \psi = \psi'')) q_\theta(\psi = \psi'') d\psi'' \right] \\
&\leq -H_\lambda - \mathrm{E}_{\psi' \sim p(\psi|\mathcal{D}, \beta)} \left[ \log q_\theta(\psi = \psi') \right] \\
&= -H_\lambda + H_\psi + \mathrm{KL}(p(\psi|\mathcal{D}, \beta)||q_\theta(\psi)) \\
&= c + \mathrm{KL}(p(\psi|\mathcal{D}, \beta)||q_\theta(\psi))
\end{aligned}$$

(43)

where

$$\begin{aligned}
H_\lambda &= -\int p(\lambda|\mathcal{D}, x, \beta) \log p(\lambda|\mathcal{D}, x, \beta) d\lambda \\
H_\psi &= -\int p(\psi|\mathcal{D}, \beta) \log p(\psi|\mathcal{D}, \beta) d\psi
\end{aligned}$$

(44)

It is worth noting that the equality is satisfied if $f$ is an invertible function. In that case,

$$\begin{aligned}
&\int \delta(f(x, \psi = \psi') = f(x, \psi = \psi'')) q_\theta(\psi = \psi'') d\psi'' \\
&= q_\theta(\psi = \psi')
\end{aligned}$$

(45)

$\square$

### B.4.3 Proposition 4.1.3:

*Under certain conditions (i.e., $||p(y|x, \mathcal{D}, \beta) - p(y|x, \theta)||_1 \leq \frac{1}{2}$ and $\sqrt{2 \log 2 \cdot d} \leq \frac{C}{e}$), the following error bounds demonstrate the distance between the uncertainties estimated by a Dirichlet network in Eq. (10)*

*and their exact measurements from a BNN in Eq. (4):*

$$\begin{aligned}
|U_t - U_t^*| &= |\mathrm{H}[p(y|x, \theta)] - \mathrm{H}[p(y|x, \mathcal{D}, \beta)]| \\
&\leq -\sqrt{2 \log 2 \cdot d} \log \frac{\sqrt{2 \log 2 \cdot d}}{C} \\
|U_a - U_a^*| &= \left| \mathrm{E}_{p(\lambda|x, \theta)}[p(y|\lambda)] - \mathrm{E}_{p(\psi|x, \mathcal{D}, \beta)}\{\mathrm{H}[p(y|x, \psi)]\} \right| \\
&\leq \log C \sqrt{2 \log 2 \cdot d} \\
|U_t - U_t^*| &= |\mathrm{I}[y, \lambda|x, \theta] - \mathrm{I}[y, \psi|x, \mathcal{D}, \beta]| \\
&\leq \sqrt{2 \log 2 \cdot d}(2 \log C - \log \sqrt{2 \log 2 \cdot d})
\end{aligned}$$

(46)

*where $d = KL(p(\lambda|\mathcal{D}, x, \beta)||p(\lambda|x, \theta))$.*

*Proof.* We first introduce a Lemma for the L1 bound on the entropy function.

**Lemma B.1** (Theorem 17.3.3 in Cover and Thomas [2006])**.** *Let $p$ and $q$ be two probability mass functions on $\mathcal{X}$ such that*

$$||p - q||_1 = \sum_{x \in \mathcal{X}} |p(x) - q(x)| \leq \frac{1}{2}.$$ (47)

*Denote $\mathrm{H}$ as the entropy function, we can obtain*

$$|\mathrm{H}(p) - \mathrm{H}(q)| \leq -||p - q||_1 \log \frac{||p - q||_1}{|\mathcal{X}|}$$ (48)

*where $|\mathcal{X}|$ is the number of elements in $\mathcal{X}$.*

**Lemma B.2.** $-||p - q||_1 \log \frac{||p - q||_1}{|\mathcal{X}|}$ *is non-decreasing with respect to $||p - q||_1$ if $||p - q||_1 \leq \frac{|\mathcal{X}|}{e}$.*

*Proof.* Let $g(t) = -t \log \frac{t}{|\mathcal{X}|}$. This is because $g'(t) = -1 - \log \frac{t}{|\mathcal{X}|} \geq 0$ is always true for $t \leq \frac{|\mathcal{X}|}{e}$. $\square$

**Total uncertainty error bound** Given the expression of the total uncertainty measured by a BNN, i.e., $\mathrm{H}[p(y|x, \mathcal{D}, \beta)]$, and its approximation by the Dirichlet network, i.e., $\mathrm{H}[p(y|x, \theta)]$, the uncertainty error bound is shown in Eq. (49) by substituting $p = p(y|x, \mathcal{D}, \beta)$ and $q = p(y|x, \theta)$ in Eq. (48).

$$\begin{aligned}
&|\mathrm{H}(p(y|x, \mathcal{D}, \beta)) - \mathrm{H}(p(y|x, \theta))| \\
&\leq -||p(y|x, \mathcal{D}, \beta) - p(y|x, \theta)||_1 \log \frac{||p(y|x, \mathcal{D}, \beta) - p(y|x, \theta)||_1}{C}
\end{aligned}$$

(49)

Then, $||p(y|x, \mathcal{D}, \beta - p(y|x, \theta)||_1$ can be expressed as:

$$
\begin{aligned}
&||p(y|x, \mathcal{D}, \beta) - p(y|x, \theta)||_1 \\
&= \sum_{k=1}^{C} |p(y = k|x, \mathcal{D}, \beta) - p(y = k|x, \theta)| \\
&= \sum_{k=1}^{C} |\int p(y = k|\lambda)p(\lambda|x, \mathcal{D}, \beta)d\lambda \\
&\quad - \int p(y = k|\lambda)p(\lambda|x, \theta)d\lambda| \\
&= \sum_{k=1}^{C} \left| \int p(y = k|\lambda) \left[ p(\lambda|x, \mathcal{D}, \beta) - p(\lambda|x, \theta) \right] d\lambda \right| \\
&\leq \sum_{k=1}^{C} \int p(y = k|\lambda) |p(\lambda|x, \mathcal{D}, \beta) - p(\lambda|x, \theta)| \, d\lambda \\
&= \int \left[ \sum_{k=1}^{C} p(y = k|\lambda) \right] |p(\lambda|x, \mathcal{D}, \beta) - p(\lambda|x, \theta)| \, d\lambda \\
&= \int |p(\lambda|x, \mathcal{D}, \beta) - p(\lambda|x, \theta)| \, d\lambda \\
&= ||p(\lambda|x, \mathcal{D}, \beta) - p(\lambda|x, \theta)||_1
\end{aligned}
\tag{50}
$$

By Prinsker's inequality [Canonne, 2022], we have

$$
\begin{aligned}
&||p(\lambda|x, \mathcal{D}, \beta) - p(\lambda|x, \theta)||_1 \\
&\leq \sqrt{2 \log 2 \text{KL}(p(\lambda|\mathcal{D}, x, \beta) || p(\lambda|x, \theta))} \\
&= \sqrt{2 \log 2 \cdot d}.
\end{aligned}
\tag{51}
$$

According to Lemma B.2, by substituting into Eq. (50) and Eq. (51) into Eq. (49), we can obtain:

$$
\begin{aligned}
&|\text{H}[p(y|x, \theta)] - \text{H}[p(y|x, \mathcal{D}, \beta)]| \\
&\leq -\sqrt{2 \log 2 \cdot d} \log \frac{\sqrt{2 \log 2 \cdot d}}{C}
\end{aligned}
\tag{52}
$$

**Aleatoric uncertainty error bound** Given the aleatoric uncertainty measured by a BNN, i.e., $\text{E}_{p(\psi|\mathcal{D}, \beta)}\{\text{H}[p(y|x, \psi)]\}$ and the aleatoric uncertainty quantified by the Dirichlet model, i.e., $\text{E}_{p(\lambda|x, \theta)}[p(y|\lambda)]$,

the aleatoric uncertainty error bound is shown as below:

$$
\begin{aligned}
&\left| \text{E}_{p(\lambda|x, \theta)}\{\text{H}[p(y|\lambda)]\} - \text{E}_{p(\psi|\mathcal{D}, \beta)}\{\text{H}[p(y|x, \psi)]\} \right| \\
&= \left| \text{E}_{p(\lambda|x, \theta)}\{\text{H}[p(y|\lambda)]\} - \text{E}_{p(\lambda|\mathcal{D}, \beta)}\{\text{H}[p(y|\lambda)]\} \right| \\
&= | \int \left[ -\sum_{k=1}^{C} p(y = k|\lambda) \log \frac{1}{p(y = k|\lambda)} \right] \\
&\quad \cdot [p(\lambda|x, \theta) - p(\lambda|x, \mathcal{D}, \beta)]d\lambda| \\
&\leq \int \left[ -\sum_{k=1}^{C} p(y = k|\lambda) \log \frac{1}{p(y = k|\lambda)} \right] \\
&\quad \cdot |p(\lambda|x, \theta) - p(\lambda|x, \mathcal{D}, \beta)|d\lambda \\
&\leq \int \log C |p(\lambda|x, \theta) - p(\lambda|x, \mathcal{D}, \beta)|d\lambda \\
&= \log C \cdot ||p(\lambda|x, \mathcal{D}, \beta) - p(\lambda|x, \theta)||_1 \\
&\leq \log C \sqrt{2 \log 2 \cdot d}
\end{aligned}
\tag{53}
$$

In Eq. (53), $\left[ -\sum_{k=1}^{C} p(y = k|\lambda) \log \frac{1}{p(y=k|\lambda)} \right] \leq \log C$ can be proved by a special property that uniform distribution has the maximum entropy.

**Epistemic uncertainty error bound** Given the epistemic uncertainty captured by $\text{I}[y; \lambda|x, \theta]$ and $\text{I}[y; \psi|x, \mathcal{D}, \beta]$ for a BNN and a Dirichlet-based model respectively,

$$
\begin{aligned}
&| \text{I}[y; \lambda|x, \theta] - \text{I}[y; \psi|x, \mathcal{D}, \beta]| \\
&= \left| \left[ \text{H}(p(y|x, \theta)) - \text{E}_{p(\lambda|x, \theta)}\{\text{H}[p(y|\lambda)]\} \right] \right. \\
&\quad + \left. \left[ \text{H}(p(y|x, \mathcal{D}, \beta)) - \text{E}_{p(\psi|\mathcal{D}, \beta)}\{\text{H}[p(y|x, \psi)]\} \right] \right| \\
&\leq |\text{H}(p(y|x, \mathcal{D}, \beta)) - \text{H}(p(y|x, \theta))| \\
&\quad + \left| \text{E}_{p(\lambda|x, \theta)}\{\text{H}[p(y|\lambda)]\} - \text{E}_{p(\psi|\mathcal{D}, \beta)}\{\text{H}[p(y|x, \psi)]\} \right| \\
&\leq -\sqrt{2 \log 2 \cdot d} \log \frac{\sqrt{2 \log 2 \cdot d}}{C} + \log C \sqrt{2 \log 2 \cdot d} \\
&\leq 2 \log C \sqrt{2 \log 2 \cdot d} - \sqrt{2 \log 2 \cdot d} \log \sqrt{2 \log 2 \cdot d}
\end{aligned}
\tag{54}
$$

**Assumptions** It is worth noting that the uncertainty error bounds are valid given the following conditions:

$$
\begin{aligned}
&||p(y|x, \mathcal{D}, \beta) - p(y|x, \theta)||_1 \leq \frac{1}{2} \\
&||p(\lambda|x, \mathcal{D}, \beta) - p(\lambda|x, \theta)||_1 \leq \sqrt{2 \log 2 \cdot d} \leq \frac{C}{e}
\end{aligned}
\tag{55}
$$

Even though we can't always assure the validity of the aforementioned assumptions, they hold true if $p(\lambda|x, \theta)$ approaches $p(\lambda|x, \mathcal{D}, \beta)$. This serves as the basis for our work, prompting us to incorporate a regularization term that minimizes the difference between $p(\lambda|x, \theta)$ and $p(\lambda|x, \mathcal{D}, \beta)$. □

## C  LA DISTILLATION LOSS

### C.1  COMPONENTS OF EQ. (18)

In Eq. (18), the detailed expression of $l(f_{map})$, $J_l$, and $H_l$ are shown in the following.

$$l(f_{map}) = \sum_{k=1}^{C} (\alpha_k - 1) \log \frac{\exp(f_k(x, \psi_{map}))}{\sum_{j=1}^{C} \exp(f_j(x, \psi_{map}))};$$

$$(56)$$

$J_l$ is the Jacobian matrix of size $C \times 1$ and

$$J_l[i] = \frac{\partial l(f)}{\partial f_i} = \alpha_i - 1 - [\sum_{k=1}^{C} (\alpha_k - 1)]\lambda_i; \quad i = 1, 2, ..., C$$

$$(57)$$

$H_l$ is the Hessian matrix of size $C \times C$ and

$$H_l[i,j] = \frac{\partial^2 l(f)}{\partial f_i \partial f_j} = \begin{cases} [\sum_{k=1}^{C} (\alpha_k - 1)]\lambda_i(\lambda_i - 1) & \text{if } i = j \\ [\sum_{k=1}^{C} (\alpha_k - 1)]\lambda_i\lambda_j & \text{if } i \neq j \end{cases}$$

$$(58)$$

### C.2  DERIVATION OF EQ. (19)

The key of obtaining Eq. (19) is to solve the expectation analytically in Eq. (9). Based on the second-order Taylor expansion shown in Eq. (18), we can obtain

$$E_{p(\psi|\mathcal{D},\beta)} \left[ \sum_{k=1}^{K} (\alpha_k - 1) \log \lambda_k(x, \psi) \right]$$

$$= E_{f \sim \mathcal{N}(f_{map}, \Sigma_f)} \left[ \sum_{k=1}^{K} (\alpha_k - 1) \log \frac{\exp(f_k)}{\sum_{j=1}^{K} \exp(f_j)} \right]$$

$$\approx E_f \left[ l(f_{map}) + J_l^T (f - f_{map}) \right.$$

$$\left. + \frac{1}{2} (f - f_{map})^T H_l (f - f_{map}) \right]$$

$$= l(f_{map}) + \frac{1}{2} tr(H_l \Sigma_f)$$

$$(59)$$

## D  DIR-BNN REFINEMENT USING NORMALIZING FLOWS

### D.1  THE DESIGN OF THE NORMALIZING FLOW

We develop the normalizing flow in accordance with these criteria:

- The transformation function, denoted as $g_\omega : \lambda \to \tilde{\lambda}$, is invertible.

- Given a simplex-valued probability vector as input, the result of $g_\omega(\cdot)$ also returns a simplex-valued probability vector.

- The calculation of the Jacobian matrix for $g_\omega$ is straightforward and manageable.

The invertible function $g_\omega = h \circ \sigma \circ \tilde{g}_\omega \circ \sigma^{-1} \circ h^{-1}$ is shown below:

$$v = h^{-1}(\lambda) : \begin{cases} v_1 = \lambda_1 \\ v_k = \frac{\lambda_k}{1 - \sum_{i=1}^{k-1} \lambda_i} & k \in [2, 3, \cdots, C-1] \end{cases}$$

$$(60)$$

$$z = \sigma^{-1}(v) : z_k = \log \frac{v_k}{1 - v_k} \quad k \in [1, 2, \cdots, C-1]$$

$$(61)$$

$$\tilde{v} = \sigma(\tilde{z}) : \tilde{v}_k = \frac{1}{1 + \exp(-\tilde{z}_k)} \quad k \in [1, 2, \cdots, C-1]$$

$$(62)$$

$$\tilde{\lambda} = h(\tilde{v}) : \begin{cases} \tilde{\lambda}_1 = \tilde{v}_1 \\ \tilde{\lambda}_k = \left( 1 - \sum_{i=1}^{k-1} \tilde{\lambda}_i \right) \tilde{v}_k & k \in [2, 3, \cdots, C-1] \\ \tilde{\lambda}_C = 1 - \sum_{i=1}^{C-1} \tilde{\lambda}_i \end{cases}$$

$$(63)$$

where $\tilde{g}_\omega$ represents a standard normalizing flow for continuous variables, the specifics of which can differ based on the method employed. For the purposes of this study, we utilize coupling flows.

### D.2  DETAILS OF EQ. (20)

In Equation (20), the expression $Dir(g_\omega^{-1}(\lambda); \alpha(x, \theta^*))$ is obtained by inserting $g_\omega^{-1}(\lambda)$ into the density function of a Dirichlet distribution, parameterized by $\alpha(x, \theta^*)$. Further elaboration on the determinant of the Jacobian matrix, specifically $\det J_{g_\omega^{-1}}(\tilde{\lambda})$, is provided in the subsequent text:

$$J_{g_\omega^{-1}}(\tilde{\lambda}) = \frac{dh^{-1}(\tilde{\lambda})}{d\tilde{\lambda}} \cdot \frac{d\sigma^{-1}(\tilde{v})}{d\tilde{v}}|_{\tilde{v}=h^{-1}(\tilde{\lambda})} \cdot \frac{d\tilde{g}_\omega^{-1}(\tilde{z})}{d\tilde{z}}|_{\tilde{z}=\sigma^{-1}(\tilde{v})}$$

$$\cdot \frac{d\sigma(z)}{dz}|_{z=\tilde{g}_\omega(\tilde{z})} \cdot \frac{dh(v)}{dv}|_{v=\sigma(z)}.$$

$$(64)$$

As a result,

$$\det J_{g_\omega^{-1}}(\tilde{\lambda}) = \det \frac{dh^{-1}(\tilde{\lambda})}{d\tilde{\lambda}} \cdot \det \frac{d\sigma^{-1}(\tilde{v})}{d\tilde{v}} \cdot \det \frac{d\tilde{g}_\omega^{-1}(\tilde{z})}{d\tilde{z}}$$

$$\cdot \det \frac{d\sigma(z)}{dz} \cdot \det \frac{dh(v)}{dv}$$

$$(65)$$

where

$$\det \frac{dh^{-1}(\tilde{\lambda})}{d\tilde{\lambda}} = \prod_{k=2}^{C-1} \frac{1}{1 - \sum_{i=1}^{k-1} \tilde{\lambda}_i}$$

$$\det \frac{d\sigma^{-1}(\tilde{v})}{d\tilde{v}} = \prod_{k=1}^{C-1} \frac{1}{\tilde{v}_k(1 - \tilde{v}_k)}$$

$$\det \frac{d\sigma(z)}{dz} = \prod_{k=1}^{C-1} \sigma(z)_k(1 - \sigma(z)_k) \quad (66)$$

$$\det \frac{dh(v)}{dv} = \prod_{k=2}^{C-1} \left(1 - \sum_{i=1}^{k-1} h(v)_i\right)$$

where $\sigma(z)_k$ and $h(v)_i$ are the kth and ith element of $\sigma(z)$ and $h(v)$. $\det \frac{d\tilde{g}_\omega^{-1}(\tilde{z})}{d\tilde{z}}$ depends on the chosen standard NF for continuous variables.

# E   EXPERIMENT SETTINGS AND IMPLEMENTATION

## E.1   MODEL ARCHITECTURE AND HYPERPARAMETERS

We conducted experiments on four datasets: MNIST, FM-NIST, C10, and C100. For all training sessions, we randomly allocate 10% of the training data as validation data for model selection. We utilize an RTX2080Ti GPU to perform all the experiments. Below, we detail the training procedures for the pre-trained models corresponding to each of the four datasets.

- MNIST/FMNIST: We employ a simple CNN architecture: Conv2D-ReLU-Conv2D-ReLU-MaxPool2D-Dense-ReLU-Dense-Softmax. Each convolutional layer includes 32 filters with a $4 \times 4$ kernel size. We utilize a max-pooling layer with a $2 \times 2$ kernel and dense layers comprising 128 units. The SGD optimizer is used with a learning rate of 1e-2 and a momentum of 0.9. We set the maximum number of epochs at 30 and the weight decay coefficient at 5e-4. The batch size is 128.

- C10/C100: We utilize ResNet18 for feature extraction, connected to a fully-connected layer for classification. The SGD optimizer is employed with an initial learning rate of 1e-1, decreasing to 1e-2, 1e-3, and 1e-4 at the 30th, 60th, and 90th epochs, respectively. The momentum is set at 0.9, with a maximum of 100 epochs and a weight decay coefficient of 5e-4. Standard data augmentation techniques, such as random cropping, horizontal flipping, and random rotation, are applied. The batch size is 128.

## E.2   IMPLEMENTATION DETAILS

In this section, we will discuss the implementation details for different uncertainty estimation methods used for OOD detection and image classification under distributional shifts.

- The Ensemble method (ESB): we randomly train 5 ensemble models with different initialization.

- PriorNet, PosNet, EviNet: we utilize implementations from Kopetzki et al. [2021] are utilized, adhering to the default hyperparameters. The source code is available at `https://github.com/TUM-DAML/dbu-robustness`. Since we do not have OOD samples, we utilize the images with random noise as the OOD samples for PriorNet.

- LA: as elaborated in Appendix A, we opt for last-layer LA accompanied by a full Hessian matrix computation. The software used is credited to Daxberger et al. [2021a] and can be found at `https://github.com/AlexImmer/Laplace`.

- DUQ: We use the source code available at `https://github.com/y0ast/deterministic-uncertainty-quantification`, modified for our experimental settings. After tuning the hyperparameters, we set the gradient penalty coefficient to 0.5 and retain other default hyperparameters.

- DUE: We utilize the open-source code available at `https://github.com/y0ast/DUE`, modified for our experimental settings. We crafted our own feature extractor using Resnet18 with spectral normalization, given that only a wide-Resnet version is accessible. While we tested various kernel types, "RBF" was our choice. We also adjusted parameters like learning rate, batch size, and dropout rate, but since these changes didn't notably affect the outcomes, we opted to stick with the default values.

- Dir-LA: we use $\rho = 50$ which is the coefficient of the regularization loss. After training a single network using MAP loss, the last-layer LA is performed with an implementation shown in `https://github.com/AlexImmer/Laplace`. Then, we add the regularization loss to refine the model until convergence.

- Dir-ESB: knowledge distillation was performed after training an ensemble of 5 models. The regularization loss is mentioned in Eq. (9), where samples from the ensemble models approximate $p(\psi \mid \mathcal{D}, \beta)$. We also select $\rho = 50$.

- Dir-ESB-NF: a 10-layer coupling flow for continuous variables was applied for $\tilde{g}_\omega$. Adhering to RealNVP [Dinh et al., 2016], the coupling flows were designed such that the affine autoregressive flow linearly scales and modifies half of the dimension as a function of the other half. The implementation of the affine flow can be found at pytorch-normalizing-flows on GitHub. We

Table 7: Uncertainty calibration within-dataset performance ( ACC (%) ↑, ALC ↑, and ROC (%) ↑ ) for MNIST and C10 datasets. "*" represents our method.

| Method | MNIST | | C10 | |
|---|---|---|---|---|
| | ACC | ALC | ACC | ALC |
| Ensemble | **99.41** | **0.984** | **93.64** | 0.902 |
| PriorNet | 99.11 | 0.983 | 64.18 | 0.670 |
| PosNet | 98.90 | 0.917 | 91.23 | 0.810 |
| EvNet | 99.18 | -1.197 | 91.77 | -0.185 |
| DUQ | 98.64 | 0.976 | 91.11 | 0.887 |
| DUE | 98.19 | 0.954 | 87.53 | 0.684 |
| LA | 98.51 | 0.959 | 92.55 | **0.915** |
| Dir-LA* | 99.34 | 0.941 | 93.16 | 0.890 |
| Dir-ESB* | 99.30 | 0.977 | 92.43 | 0.914 |
| Dir-ESB-NF* | 99.31 | **0.984** | 93.16 | 0.884 |

Table 8: Additional results for uncertainty calibration performance ( ACC (%) ↑, ALC ↑, and ROC (%) ↑ ) for MNIST. ROC represents AUROC. "*" represents our method.

| Method | Rotate 40 | | | Rotate 80 | | | Rotate 120 | | |
|---|---|---|---|---|---|---|---|---|---|
| | ACC | ALC | ROC | ACC | ALC | ROC | ACC | ALC | ROC |
| Ensemble | 71.27 | 0.666 | 88.89 | 17.34 | 0.180 | 97.13 | 20.65 | 0.168 | 96.16 |
| PriorNet | 68.81 | 0.694 | 86.84 | 15.42 | 0.292 | 96.52 | 16.89 | 0.200 | 95.76 |
| PosNet | 64.19 | **0.748** | 88.24 | 16.35 | **0.415** | 89.47 | 20.22 | 0.037 | 82.61 |
| EvNet | 71.77 | -0.124 | 32.19 | 16.66 | -0.006 | 25.86 | 22.80 | 0.035 | 33.15 |
| DUQ | 66.03 | 0.673 | 90.49 | 17.84 | 0.195 | 97.18 | 16.50 | 0.127 | 96.53 |
| DUE | 59.89 | 0.676 | 84.17 | 15.94 | 0.173 | 89.55 | 16.65 | 0.054 | 86.10 |
| LA | 66.54 | 0.649 | 86.82 | **21.25** | 0.216 | 95.02 | 19.95 | 0.123 | 93.86 |
| Dir-LA* | **73.00** | 0.710 | 90.39 | 19.94 | 0.222 | 98.01 | 21.71 | **0.259** | 95.63 |
| Dir-ESB* | 70.54 | 0.706 | 92.06 | 19.80 | 0.212 | 98.04 | 21.63 | 0.218 | 97.04 |
| Dir-ESB-NF* | 70.81 | 0.709 | 92.18 | 19.47 | 0.237 | 98.21 | 22.17 | 0.225 | 97.21 |

initialize the NF to start with the pre-trained Dirichlet distribution. During the training phase of NF, we used the Adam optimizer with a learning rate of $1 \times 10^{-4}$, and the maximum training iterations were set to 400.

# F   ADDITIONAL RESULTS FOR UNCERTAINTY CALIBRATION UNDER DISTRIBUTIONAL SHIFTS

This section first presents the within-dataset performance measured by ACC and AULC for MNIST and C10. As demonstrated in Table 7, our techniques show a slight improvement in within-dataset performance compared to both Dirichlet-based and single-network deterministic methods. Due to the page restrictions, supplementary results, which include varying degrees of distributional shifts, are provided in Tables 8 and 9. These additional results support the conclusions drawn in Section 5.2. Tables 10 and 11 detail the negative log-likelihood (NLL) and expected calibration error (ECE) outcomes for the rotated MNIST and noisy C10 datasets, respectively, under varying distributional shifts. NLL and ECE, as pivotal calibration metrics, furnish additional evaluations on aleatoric uncertainty and total uncertainty. It's important to note that DUQ and DUE are ex-

cluded because they don't provide softmax probabilities. On average, our methods (Dir-LA, Dir-ESB) demonstrate competitive performance. Notably, Dir-LA outperforms other baselines with the best average scores in both NLL and ECE for the rotated MNIST dataset. For the noisy C10 dataset, Dir-ESB achieves the top average NLL score.

Table 9: Additional results for uncertainty calibration performance ( ACC (%) ↑, ALC ↑, and ROC (%) ↑ ) for C10. ROC represents AUROC. "*" represents our method.

| Method | Noise 0.20 | | | Noise 0.25 | | |
|---|---|---|---|---|---|---|
| | ACC | ALC | ROC | ACC | ALC | ROC |
| Ensemble | 11.52 | 0.279 | 89.02 | 10.43 | 0.249 | 91.09 |
| PriorNet | 29.35 | 0.329 | 84.80 | 21.69 | 0.251 | 89.51 |
| PosNet | 10.00 | -0.062 | 91.18 | 9.98 | -0.082 | 88.62 |
| EvNet | 17.34 | 0.135 | 72.60 | 13.66 | 0.029 | 79.30 |
| DUQ | 18.16 | 0.271 | 93.19 | 13.70 | 0.128 | 91.19 |
| DUE | 21.35 | 0.146 | 63.15 | 14.52 | 0.102 | 53.13 |
| LA | 11.97 | 0.149 | **93.69** | 10.36 | 0.119 | 90.41 |
| Dir-LA* | 16.47 | 0.221 | 91.10 | 14.19 | 0.164 | 89.18 |
| Dir-ESB* | 22.43 | 0.375 | 74.94 | 19.45 | 0.289 | 78.21 |
| Dir-ESB-NF* | 47.03 | **0.536** | 86.50 | 38.64 | **0.429** | 89.62 |

Table 10: The NLL and ECE for rotated MNIST dataset under different rotation angles from 0 to 180 with a step of 20. 0 rotation represents the in-distribution performance. The bold values indicate the best performance.

| Method | NLL ↓ for Rotated MNIST Under Distributional Shifts | | | | | | | | | | |
|---|---|---|---|---|---|---|---|---|---|---|---|
| | 0 | 20 | 40 | 60 | 80 | 100 | 120 | 140 | 160 | 180 | Avg |
| PriorNet | 0.118 | 0.263 | 1.125 | 2.225 | 2.647 | 2.791 | 2.742 | 2.474 | 2.373 | 2.357 | 1.912 |
| PosNet | 0.043 | 0.187 | 1.117 | 2.637 | 3.348 | 3.512 | 3.530 | 3.158 | 3.012 | 3.167 | 2.371 |
| EvNet | 2.184 | 2.191 | 2.226 | 2.272 | **2.292** | **2.294** | **2.288** | 2.280 | 2.271 | 2.264 | 2.256 |
| Ensemble | 0.028 | **0.127** | **0.888** | 2.391 | 3.523 | 4.128 | 4.208 | 4.194 | 4.675 | 5.207 | 2.937 |
| LA | **0.027** | 0.153 | 0.996 | 2.408 | 3.435 | 4.104 | 4.019 | 3.973 | 4.428 | 5.212 | 2.876 |
| Dir-LA | 0.454 | 0.647 | 1.268 | **2.019** | 2.298 | 2.344 | 2.305 | **2.209** | **2.130** | **2.104** | **1.778** |
| Dir-ESB | 0.039 | 0.183 | 0.974 | 2.101 | 2.727 | 3.043 | 3.152 | 2.984 | 3.071 | 3.284 | 2.156 |

| Method | ECE ↓ for Rotated MNIST Under Distributional Shifts | | | | | | | | | | |
|---|---|---|---|---|---|---|---|---|---|---|---|
| | 0 | 20 | 40 | 60 | 80 | 100 | 120 | 140 | 160 | 180 | Avg |
| PriorNet | 0.088 | 0.094 | 0.060 | 0.247 | 0.351 | 0.389 | 0.383 | 0.319 | 0.318 | 0.331 | 0.258 |
| PosNet | 0.016 | 0.023 | 0.079 | 0.317 | 0.444 | 0.480 | 0.470 | 0.404 | 0.379 | 0.394 | 0.301 |
| EvNet | 0.879 | 0.844 | 0.601 | 0.240 | **0.073** | **0.063** | **0.121** | 0.186 | 0.255 | 0.298 | 0.356 |
| Ensemble | 0.012 | 0.022 | 0.080 | 0.329 | 0.467 | 0.534 | 0.516 | 0.452 | 0.436 | 0.449 | 0.330 |
| LA | **0.003** | **0.005** | 0.099 | 0.306 | 0.422 | 0.478 | 0.474 | 0.427 | 0.428 | 0.464 | 0.311 |
| Dir-LA | 0.347 | 0.401 | 0.302 | **0.059** | 0.153 | 0.164 | 0.140 | **0.090** | **0.089** | **0.112** | **0.186** |
| Dir-ESB | 0.019 | 0.047 | **0.053** | 0.220 | 0.348 | 0.395 | 0.414 | 0.384 | 0.379 | 0.399 | 0.266 |

Table 11: The NLL and ECE for noisy C10 dataset under different noise levels from 0 to 0.25 with a step of 0.05. 0 noise represents the in-distribution performance. The bold values indicate the best performance.

| Method | NLL ↓ and ECE ↓ for noisy C10 Under Distributional Shifts | | | | | | | | | | | | | |
|---|---|---|---|---|---|---|---|---|---|---|---|---|---|---|
| | 0 | | 0.05 | | 0.1 | | 0.15 | | 0.2 | | 0.25 | | Avg | |
| | NLL | ECE | NLL | ECE | NLL | ECE | NLL | ECE | NLL | ECE | NLL | ECE | NLL | ECE |
| PriorNet | 1.895 | 0.134 | 1.898 | 0.133 | **1.941** | 0.126 | **2.020** | 0.106 | **2.141** | 0.112 | **2.235** | 0.060 | 2.022 | **0.112** |
| PosNet | 0.358 | 0.043 | 1.318 | 0.162 | 2.593 | 0.348 | 3.118 | 0.474 | 3.340 | 0.522 | 3.546 | 0.557 | 2.379 | 0.351 |
| EvNet | 1.277 | 0.061 | 1.394 | 0.098 | 2.416 | 0.341 | 3.349 | 0.472 | 3.768 | 0.528 | 3.478 | 0.371 | 2.614 | 0.312 |
| Ensemble | **0.170** | **0.008** | **0.905** | **0.081** | 3.588 | 0.435 | 5.226 | 0.534 | 5.403 | 0.515 | 5.194 | 0.505 | 3.414 | 0.346 |
| LA | 0.246 | 0.019 | 1.201 | 0.132 | 4.736 | 0.426 | 6.970 | 0.493 | 7.769 | 0.560 | 7.966 | 0.641 | 4.815 | 0.379 |
| Dir-LA | 0.697 | 0.335 | 1.398 | 0.242 | 2.443 | **0.113** | 2.707 | 0.145 | 2.698 | 0.167 | 2.640 | 0.198 | 2.097 | 0.200 |
| Dir-ESB | 0.326 | 0.061 | 0.940 | 0.083 | 2.148 | 0.291 | 2.578 | 0.373 | 2.730 | 0.378 | 2.765 | 0.357 | **1.915** | 0.257 |

# G ABLATION STUDIES AND FURTHER ANALYSIS

## G.1 EFFICIENCY ANALYSIS

This section is dedicated to assessing the efficiency of the introduced methods through both theoretical and practical evidence. Let us define $M$ as the total count of parameters, $P$ as the number of parameters in the last layer, $N$ as the data size, $C$ as the class count, $S$ as the number of samples produced for LA, $K$ as the ensemble components count, and $T$ as the number of training epochs. Both the theoretical complexity analysis and the actual training and uncertainty quantification (UQ) runtimes are detailed in Table 12.

It is important to acknowledge that theoretical investigations into model complexity do not account for diverse loss functions, but rather calculate complexity solely in relation to backpropagation for a fixed set of training samples. The Dir-LA method we propose does necessitate extra time for the last-layer LA ($O(C^3 + P^3)$) during training. However, Dir-LA doesn't significantly extend training time. As this process is executed just once throughout the entire training period, its time consumption is marginal in comparison to the overall training process. Empirically, building the last-layer LA for a single network consumes 3.2s for MNIST and 14.4s for C10.

Dir-ESB requires pre-trained ensemble models. DUQ employs the gradient penalty during training, thereby demanding an additional $O(TNM)$ for backpropagating through gradients. DUE takes advantage of spectral normalization which adds another $O(TM)$ to the training time. In practice, PriorNet doubles the training duration as it also has to load the OOD data samples for regularization. Concurrently, PosNet demands a longer duration to train the normalizing flow.

Regarding inference, most methods, except Ensemble and LA, can estimate uncertainty via a single deterministic forward pass of the neural network, thus reducing both the computational complexity and the memory requirements. The theoretical and practical runtimes of UQ for Dir-LA and Dir-ESB are comparable to other deterministic UQ approaches.

The aforementioned comparisons do not include Dir-ESB-NF as it is not part of the Dirichlet-based networks. Dir-ESB-NF implements a normalizing flow in a post-processing approach to refine the Dirichlet distribution to better mimic the posterior. Given that the NF is initialized to align with the pre-trained Dirichlet distribution, typically only a few iterations (less than 200) are necessary to learn the NF. The empirical runtime for a single iteration is 0.27s for the C10 dataset. Overall, Dir-ESB-NF is unquestionably more complex than a standard Dirichlet-based model. However, it offers alternative pathways to further enhance the Dirichlet

network with improved performance. For inference, Dir-ESB-NF requires inputting the samples from the Dirichlet distribution into the NF. The complexity here relies on the number of samples produced and the intricacy of the NF. Practically, if we generate 100 samples and only employ a shallow flow, the runtime will be 6.2s for estimating the uncertainty for the C10 testing dataset.

## G.2 MORE ANALYSIS FOR DIR-ESB REFINEMENT USING NF

**Posterior approximation** The NF we propose offers both theoretical and empirical evidence that a better approximation of the posterior distribution can be achieved. Importantly, the NF is designed to initialize with the pre-trained Dirichlet distribution, thereby ensuring that minimizing the KL divergence between the NF distribution and the posterior leads to an enhanced approximation of the posterior. Considering the substantial efforts and time needed to obtain an exact posterior distribution, this work opts for ensemble methods and LA as substitutes for BNN regularization of NF, in line with the Dir-BNN training strategy detailed in Sec. 4.1. The primary aim of training the NF is to distill vital information about the weight posterior as much as possible, without the need for an exact posterior. However, a perfect alignment with the approximate BNN is not desirable, as it could potentially hinder performance. For instance, in numerous experiments, Dir-LA demonstrates superior performance in OOD detection compared to LA.

**Choose Dir-ESB for refinement** We choose to refine Dir-ESB using NF instead of Dir-LA. This decision allows us to maintain the essential information that is intrinsic to LA, while circumventing the risk of the NF-created distribution aligning too closely with LA, considering LA serves as a strong assumption for posterior approximation.

**A shallow NF** In this study, we only require a shallow flow to refine the Dirichlet distribution. This approach serves two purposes. Firstly, it enhances efficiency. Secondly, a shallow NF suffices for refining the model. As previously mentioned, we do not aim to achieve a perfect alignment with the approximate inference methods used for distillation. Therefore, a shallow flow is adequate to glean critical information from the approximate posterior while steering clear of overfitting it. Empirically, we evaluate the OOD performance of Dir-ESB-NF by varying the number of flows. Upon increasing the complexity of the NF, there are no notable improvements.

**Few training iterations** Since our objective isn't to achieve a perfect alignment of the NF with the approximate posterior, we only need a few training iterations. The stopping criteria are dictated by the performance metrics on the validation dataset, such as accuracy and negative log-likelihood loss.

**Additional results for refining weak Dir-ESB models**

Table 12: The theoretical complexity shows the additional runtimes of each method for both training and UQ compared to a single network's training and evaluation. The empirical results of training are the runtimes of training C10 for one epoch in seconds. The results for inference show the runtimes for estimating the uncertainty of the C10 testing dataset.

| Method | Theoretical Complexity | | Empirical Runtime (C10) | |
|---|---|---|---|---|
| | Training | UQ | Training | UQ |
| Ensemble | $O((K-1)TNM)$ | $O((K-1)NM)$ | 202.2s | 5.3s |
| PriorNet | $O(1)$ | $O(1)$ | 98.4s | 1.4s |
| PosNet | $> O(1)$ | $O(1)$ | 533.0s | 1.4s |
| EvNet | $O(1)$ | $O(1)$ | 23.27s | 1.4s |
| DUQ | $O(TNM)$ | $O(1)$ | 59.5s | 1.5s |
| DUE | $O(TM)$ | $O(1)$ | 52.4s | 1.6s |
| LA | $O(C^3 + P^3)$ | $O(SNP)$ | 50.5s | 3.6s |
| Dir-LA* | $O(C^3 + P^3)$ | $O(1)$ | 59.2s | 1.4s |
| Dir-ESB* | $O(KTNM)$ | $O(1)$ | 256.6s | 1.4s |

Table 13: OOD detection results for AUROC (%) ↑ and AUPR (%) ↑ for refining a weak Dir-ESB model using Dir-ESB-NF. We also consider different ensemble sizes. "→" shows the improvement.

| Method | MNIST → KMNIST | | MNIST → FMNIST | |
|---|---|---|---|---|
| | AUROC | AUPR | AUROC | AUPR |
| Ensemble-5 | 97.8 | 97.8 | 97.4 | 97.2 |
| Dir-ESB → Dir-ESB-NF | 96.1→96.7 | 95.0→96.2 | 98.7→98.7 | 98.4→98.5 |

| Method | MNIST → KMNIST | | MNIST → FMNIST | |
|---|---|---|---|---|
| | AUROC | AUPR | AUROC | AUPR |
| Ensemble-10 | 98.6 | 98.6 | 97.9 | 97.5 |
| Dir-ESB → Dir-ESB-NF | 94.4→97.3 | 92.7→96.9 | 97.7→98.7 | 96.5→98.5 |

| Method | MNIST → KMNIST | | MNIST → FMNIST | |
|---|---|---|---|---|
| | AUROC | AUPR | AUROC | AUPR |
| Ensemble-20 | 98.6 | 98.6 | 98.0 | 97.7 |
| Dir-ESB → Dir-ESB-NF | 97.3→98.0 | 97.2→98.0 | 99.2→99.3 | 99.1→99.3 |

**with varying ensemble size** In Table 13, we provide empirical evidence of further enhancements when we employ NF to refine weak Dir-ESB models. These weak Dir-ESB models are derived by using a small coefficient $\rho$, specifically $\rho = 10$, in Eq. (8) during training. The results indicate that NF can bring about more improvements on a weak Dirichlet-based network. Additionally, it is feasible to use NF to refine Dirichlet-based networks trained using other existing methodologies.