# OpenReview forum: "Beyond Dirichlet-based Models: When Bayesian Neural Networks Meet Evidential Deep Learning"
_auai.org/UAI/2024/Conference — UAI 2024 poster_

### Official Review · Reviewer_xjs6 · 2024-03-19

**Q2-1 Originality-Novelty:** 3
**Q2-2 Correctness-Technical Quality:** 3
**Q2-5 Clarity Of Writing:** 3

**Q1 Summary And Contributions:**

This paper proposes a Dir-BNN to bridge the BNN and evidential deep learning for efficient uncertainty estimation. The idea is to add a KL divergence between the parameter posterior distribution from evidential deep learning and BNNs as a regularizer to borrow the knowledge from BNNs for uncertainty estimation. An LA-based approximation is proposed to replace the exact posterior from BNNs, named 'self-distillation'. Finally, normalizing flow is used to further extend the expressiveness of Dirichlet distribution.

**Q2-3 Extent To Which Claims Are Supported By Evidence:**

3: Good: the main claims are supported by convincing evidence (in the form of adequate experimental evaluation, proofs, (pseudo-)code, references, assumptions).

**Q2-4 Reproducibility:**

3: Good: key resources (e.g. proofs, code, data) are available and key details (e.g. proofs, experimental setup) are sufficiently well-described for competent researchers to confidently reproduce the main results.

**Q3 Main Strengths:**

1. The proposed idea of using BNNs to regularize the distribution learned from Dir-MAP is simple but effective.

2. The quality of uncertainty estimation is theoretically guaranteed.

3. The experimental results proof the effectiveness of the proposed idea.

**Q4 Main Weakness:**

1. The authors claim that "with a suboptimal BNN, adding NLL loss may improve Dir-BNN's predictions and ensure stable training" on Page 3. However, this claim is not proven. What is the reason for that?  If you have sufficient training time for BNN, you do not need Dir-MAP so we assume you only do light training for BNN. Is that possible the lightly trained BNN may hurt the performance?

2. The uncertainty error bound given in Proposition 4.1.3 uses the one from BNNs as the reference. The bound is mainly determined by the KL divergence. There is no consideration of the error from BNN "light/insufficient training".

**Q5 Detailed Comments To The Authors:**

The authors claim that "with a suboptimal BNN, adding NLL loss may improve Dir-BNN's predictions and ensure stable training" on Page 3. However, this claim is not proven. What is the reason for that?  If you have sufficient training time for BNN, you do not need Dir-MAP so we assume you only do light training for BNN. Is it possible that the lightly trained BNN may hurt the performance?

For the normalizing flow, is that possible to train the normalizing flow together with the Dir-BNN? Why is the post-processing fashion adopted?

The results in tables 1 and 2 are not accompanied by statistically significant tests. Some of them look not statistically significant.

**Q9 Complying With Reviewing Instructions:**

Yes

---

> ### Author Rebuttal · Authors · 2024-04-07
>
> Thank you so much for the detailed review. Please find our responses below.
>
> **Claims on Page 3 for Prediction Improvement**: Since approximate BNNs may misclassify some input data, sole reliance on them might prevent the evidential model from correcting prediction errors. In addition to the hyperparameter sensitivity analysis in Sec. 5.3, we will include further analysis showing that when $\rho = 100$, where Bayesian regularization is dominant and NLL loss contributes minimally, the OOD performance for MNIST $\rightarrow$ Omniglot decreases to 97.4/97.3 for AUROC/AUPR. This is lower than the performance observed with $\rho = 30$, highlighting the potential benefit of incorporating NLL loss to improve performance.
>
> **Claims on Page 3 for Stable Training**: As noted in Sec.3.1 of [1], Dirichlet-based methods may encounter optimization challenges. Optimizing Eq. 9 at the start of training is difficult because the Bayesian posterior for the output space is often "sharp" at one corner of the simplex. Initially, when the NN is randomly initialized, the Dirichlet distribution's mode is near the center, making the KL optimization challenging due to limited common support between the two distributions. Adding the NLL loss can guide the optimization in the right direction at the beginning of the training.
>
> [1] Ensemble distribution distillation.
>
> **Lightly Trained BNNs**: We acknowledge the necessity for effective BNN approximations. We avoid using poorly approximated BNNs as supervision. While obtaining good approximations of the parameter-space posterior is hard, acquiring a low-dimensional output-space posterior is easier, where ensemble methods and LA can offer reliable UQ. Good empirical performance can be achieved with these approximations. To minimize approximation errors from BNN to Dir-BNN, we integrate NLL loss with KL loss in Dir-BNN training (Eq. 8). Sometimes, Dir-BNN may outperform teacher models by distilling essential information for output-space uncertainty and omitting the irrelevant, as discussed in Sec. 5.1.
>
> **Light-trained BNN for Proposition 4.1.3**: Proposition 4.1.3 demonstrates that the estimated uncertainty by the Dirichlet-based network can closely match the exact measures if the Dirichlet distribution is optimized to be close to the Bayesian posterior. In practice, when the BNN posterior $p(\lambda|x,\mathcal{D},\beta)$ is intractable and approximated by $q(\lambda|x)$, denoting $U_t^{q}$ as the total uncertainty estimated using $q(\lambda|x)$, we have
> $$
> \left|U_t-U_t^*\right| \leq  \left|U_t-U_t^q\right| + \left|U_t^q-U_t^*\right| \leq \left|U_t^q-U_t^*\right| - \sqrt{2\log2 \cdot d} \log \frac{\sqrt{2\log2 \cdot d}}{C}
> $$
> Where $d = \text{KL}(q(\lambda|x)||p(\lambda|x,\theta))$. Notably, this bound accounts for errors from the Bayesian approximation but may not be tight. We emphasize that an effective approximation for the BNN, such as an ensemble or LA method, is often sufficient to achieve good empirical performance.
>
> **Training NF**: It's feasible to simultaneously train Dir-BNN to learn $\omega, \theta$ together by using the loss function in Eq. 8 and substituting Eq. 9's regularization with Eq.19. We opted for post-processing because (1) Training together might undermine the Dirichlet distribution's role as a conjugate prior, but in our scenario, it remains effective for UQ and quick inference.  The post-processing step can serve as an alternative strategy for improving UQ accuracy, and we can still use the Dirichlet distribution for fast inference. (2) Starting from scratch poses challenges, as learning Dirichlet and NF distributions together is generally harder. Using NF for post-processing requires only shallow flows. (3) This approach ensures the NF distribution more closely aligns with the Bayesian posterior than the Dirichlet distribution. It may avoid potential local minima and improve performance.
>
> **Statistical Test**:  We will present the statistical tests for Table 1 here and include the remaining results in the revised paper. We performed a Wilcoxon signed-rank test to test the hypothesis that Dir-LA, Dir-ESB, and Dir-ESB-NF have larger AUROC/AUPR scores. The Wilcoxon signed-rank test is chosen because (1) Scores are paired based on datasets; (2) the normality assumption does not hold, requiring a non-parametric test. The pairwise p-values are shown below. A p-value <0.05 indicates that our method is statistically better than the corresponding baseline.
>
> Method |PriorNet|PosNet|EviNet|DUQ|DUE|LA| ESB|
> ---|---|---|---|---|---|---|---|
> Dir-LA |<0.01|<0.01|<0.01|<0.01| <0.01|0.022| 0.510|
> Dir-ESB |0.018|<0.01|<0.01|0.017| <0.01|0.037| 0.698|
> Dir-ESB-NF |<0.01|<0.01|<0.01|<0.01| <0.01|<0.01| <0.01|
>
> We can see that Dir-LA and Dir-ESB are statistically better than others except ESB. However, Dir-LA and Dir-ESB leverage single, sampling-free evidential networks that enable fast UQ, whereas ESB relies on multiple models and requires multiple forward passes for UQ.

---

### Official Review · Reviewer_mWy3 · 2024-03-20

**Q2-1 Originality-Novelty:** 3
**Q2-2 Correctness-Technical Quality:** 3
**Q2-5 Clarity Of Writing:** 4

**Q1 Summary And Contributions:**

The paper proposes the approach for Bayesian knowledge distillation into an evidential network. Two important improvements are: a single forward pass, that is achieved with Laplacian approximation and a refinement with normalizing flows that transform from Dirichlet into more expressive posterior distribution. The experiments show overall good performance for out-of-distribution detection.

**Q2-3 Extent To Which Claims Are Supported By Evidence:**

3: Good: the main claims are supported by convincing evidence (in the form of adequate experimental evaluation, proofs, (pseudo-)code, references, assumptions).

**Q2-4 Reproducibility:**

3: Good: key resources (e.g. proofs, code, data) are available and key details (e.g. proofs, experimental setup) are sufficiently well-described for competent researchers to confidently reproduce the main results.

**Q3 Main Strengths:**

I really enjoyed reading the paper, unfortunately due to short time and length of the content, which is dense and split into appendix, I can't fully dive into paper and check all the intricate details. Overall, paper appears to be technically sound.

I believe that the novelty is good, as there are multiple non-trivial contributions, that are also expansively described in the appendix. Having the efficient methods of Bayesian knowledge distillation or learning the evidential models would be a significant contribution to the area.

The paper provides a lot of appendices with proofs and theoretical estimates.

**Q4 Main Weakness:**

Unfortunately, I think that experiments are small scale and even for this scale of experiments the results are overall only incrementally improved compared to baselines.

Code was not provided, so it is difficult to evaluate the implementation complexity of the method.

**Q5 Detailed Comments To The Authors:**

Minor things:
* Ablation: Hyperparameter sensitivity: very vaguely written
* Conclusion: to relax the BNN assumption - which one? this is not a self-sufficient text
* Badly formatted links in related works
* Different abstract in openreview and paper is really confusing

Questions:
* Can you describe the resulting distribution after NF? How far is it from Dirichlet posterior?
* One of the mentioned advantages is that the method does not require sampling, is there a time complexity, or concrete running time comparsion for this difference?

**Q9 Complying With Reviewing Instructions:**

Yes

---

> ### Author Rebuttal · Authors · 2024-04-07
>
> Thank you for the thoughtful comments. Please take a look at our responses below.
>
> **Small-scale Experiments**: We appreciate the suggestion to extend the proposed method to more complex datasets to demonstrate its effectiveness. We have observed applications of evidential deep learning in open-set recognition tasks, such as in [2], where more complex datasets are evaluated. This shows that our proposed work can be adapted to complex vision tasks. Applying our algorithm to more intricate computer vision tasks will be the focus of our future work. We also want to highlight that we evaluated our method on benchmark image classification datasets commonly used for Dirichlet-based evidential networks. Appendix B’s Table 5 in Ulmer. 2021 [1] provides a summary of the datasets utilized in this field. The datasets we used are comparable to those in existing works.
>
> [1] A survey on evidential deep learning for single-pass uncertainty estimation.
>
> [2] Evidential Deep Learning for Open Set Action Recognition
>
> **Incremental Improvement**: To clarify, compared to other Dirichlet-based methods such as PriorNet, PostNet, and EviNet, our method demonstrates a more significant improvement. When compared with single-network UQ methods like DUQ and DDU, our method also shows noticeable improvement. In comparison to ensemble methods, we achieve only comparable results. However, our method leverages single, sampling-free evidential networks that enable fast UQ. Ensemble methods, on the other hand, rely on multiple models and require multiple forward passes for UQ.
>
> **Implementation Complexity**: We plan to release the code after publication. Implementing the proposed method is generally straightforward. Firstly, obtaining a BNN for supervision is not challenging, as various methods can be utilized. The Dirichlet-based network, being a single network, is easy to implement. Furthermore, implementing the loss function described in Eq. 8 for training the Dirichlet network is also not difficult.
>
> **NF Distribution**: The resulting distribution from NF does not conform to the common distributions we are familiar with, making it challenging to directly derive an easily understandable density function. This complexity is often considered the beauty of NF, as it can model complex distributions by leveraging NNs. Instead of directly assessing how it deviates from the Dirichlet posterior, our interest lies more in its proximity to the true Bayesian posterior. By directly optimizing Eq. 19 and starting from the Dirichlet posterior, the NF distribution is guaranteed to be closer to the true Bayesian posterior. We can often observe that the KL divergence between the NF distribution and the Bayesian posterior can be reduced by 1/3 throughout training. This also suggests that the NF distribution is distinctly different from the Dirichlet posterior.
>
> **Time Complexity**: The theoretical time complexity and empirical runtime for both training and UQ are detailed in Appendix G.1. We will highlight this analysis in the main body of the paper.
>
> **Minor Issues**: For the hyperparameter sensitivity study, we will revise this section to ensure clear communication of: (1) the intuitions behind selecting the regularization coefficient, and (2) a detailed analysis of the experimental results. In the conclusion section, the BNN assumption refers to the fact that the Dir-BNN method requires a pre-trained BNN for supervision. Dir-LA integrates BNN training into Dirichlet network training using LA, thereby relaxing the need for a pre-trained BNN through self-distillation. These clarifications will be made explicit in the revised paper. Additionally, we will meticulously format the links in the related works and update the OpenReview abstract to reflect the latest version.

---

### Official Review · Reviewer_75vQ · 2024-03-23

**Q2-1 Originality-Novelty:** 3
**Q2-2 Correctness-Technical Quality:** 3
**Q2-5 Clarity Of Writing:** 3

**Q1 Summary And Contributions:**

The paper proposes enhancing Dirichlet-based evidential deep learning with Bayesian neural networks distillation in training (Dir-BNN). Three practical implementations for this method are also presented: self-distillation with Laplace approximation (Dir-LA), distilling ensemble-approximated BNN (Dir-ESB), and Dir-ESB with normalizing flow (NF) refinement.

**Q2-3 Extent To Which Claims Are Supported By Evidence:**

4: Excellent: all claims are supported by very convincing evidence (in the form of comprehensive experimental evaluation, rigorous mathematical proofs, detailed (pseudo-)code, precise references, well-motivated and realistic assumptions) and the authors deliver what they promise.

**Q2-4 Reproducibility:**

3: Good: key resources (e.g. proofs, code, data) are available and key details (e.g. proofs, experimental setup) are sufficiently well-described for competent researchers to confidently reproduce the main results.

**Q3 Main Strengths:**

- The paper presents a novel approach to improve evidential deep learning with well-calibrated BNNs, while still retain the efficiency of evidential deep learning during inference.
- The paper presents a fairly comprehensive development of the framework and implementations, with appropriate technical insights and guarantees.
- Experiments are comprehensive with performance on the practical OOD detection task, uncertainty calibration analysis, and ablation studies.
- Experimental results successfully demonstrate state-of-the-art performance of Dir-BNN approaches.

**Q4 Main Weakness:**

- The relationship between proposed Dir-BNN and the practical Dir-ESB are not clearly presented. Dir-ESB also does not have sufficient implementation details.
- While Dir-BNN seems to be compatible with many BNN approximation, only last-layer LA and ensembles are evaluated.
- Algorithm 1 seems to describe only the computation of the training loss, not the training procedure.

**Q5 Detailed Comments To The Authors:**

Although this is an interesting paper, I believe its presentation could be improved from the points in Q4. Additionally,

- “HPNN” in Figure 1 is not defined in the text,
- “Refinement” is used interchangeably with regularization during training in the context of LA (especially in Section 4.2), while the rest of the text refer to refinement as a post-hoc process.

I suggest the author address these points prior to publication.

**Q9 Complying With Reviewing Instructions:**

Yes

---

> ### Author Rebuttal · Authors · 2024-04-07
>
> Thank you so much for the detailed comments. Please take a look at our responses below.
>
> **Dir-BNN and Dir-ESB**: Dir-BNN is trained using Eq. 8 given a BNN $p(\psi|\mathcal{D},\beta)$. Since the exact posterior distribution is intractable, in practice, we can approximate it using $S$ samples $\psi^{1}, …,\psi^{S}$ through the ensemble method. To train Dir-ESB, we replace the expectation term in Eq. 8 with a sample average:
>
> $$
> E_{p(\psi|\mathcal{D},\beta)} \left[ \sum_{k=1}^{C}(\alpha_{k}-1) \log \lambda_{k}(x,\psi) \right ] \rightarrow \frac{1}{S} \sum_{s=1}^S \sum_{k=1}^{C}\left(\alpha_{k}-1\right) \log \lambda_{k}(x,\psi^s)
> $$
>
> **Implementation of Dir-ESB**: While the loss function of Dir-ESB has been discussed above, we will provide more implementation details, including maximum epoch, learning rate, coefficient $\rho$, batch size, etc., in the revised paper. For instance, for the MNIST dataset, we essentially adhere to the implementation settings outlined in Appendix E.1, training the Dir-ESB with the same hyperparameters and selecting $\rho=50$. We will clarify these details for all datasets in the revised paper.
>
> **Other BNNs for Supervision**: Dir-BNN is compatible with many BNN approximations. We chose the ensemble and LA methods for their effectiveness in UQ. To further demonstrate Dir-BNN's effectiveness, we are exploring its combination with SGLD [1] and SGHMC [2]. MCMC methods are powerful techniques to approximate the posterior distribution if the burn-in period is long enough and sufficient samples are generated. For implementing SGLD and SGHMC, we followed the experimental settings outlined in Appendix E.1 to ensure consistency in model architecture and certain training protocols. For the burn-in period, we used 50 epochs. After the burn-in period, we selected 50 samples from each method as supervision for the Dirichlet-based model. The OOD detection performance (AUROC/AUPR) on MNIST and FMNIST datasets is shown below:
>
> Method | MNIST -> Omniglot | MNIST -> KMNIST| FMNIST -> EMNIST | FMNIST -> MNIST|
> ---|---|---|---|---|
> Dir-ESB |97.7/97.6|98.5/98.3|92.0/93.3|92.0/92.3|
> Dir-SGLD |97.9/97.3|98.4/98.3|94.1/97.3|94.6/95.7|
> Dir-SGHMC |98.0/98.0|97.4/97.3|90.4/95.7|91.5/93.5|
>
> It is shown that Dir-SGLD and Dir-SGHMC also perform competitively. More analyses and implementation details will be provided in the revised paper.
>
> [1] Bayesian learning via stochastic gradient Langevin dynamics.
>
> [2] Stochastic gradient hamiltonian monte carlo.
>
> **Algorithm 1**: Due to the page limit, we primarily present the training steps in Algorithm 1 by showcasing the loss functions, as the loss function is the most crucial part of the training. In the revision of Algorithm 1, we plan to explicitly detail the iterative training procedure. This includes incorporating steps for parameter updates, monitoring evaluation metrics, and conducting convergence checks to ensure a comprehensive depiction of the training process. It will involve outlining the initialization of model parameters, detailing the sequence of operations for each training epoch, and specifying the criteria for model optimization and regularization.
>
> **Presentation**: We apologize for the typo in Figure 1 and will correct 'HPNN' to Dir-BNN. We will also revise the text to ensure that 'refinement' exclusively refers to the post-hoc process. In all other contexts, we will use 'regularization'.

---

### Official Review · Reviewer_BhS5 · 2024-03-26

**Q2-1 Originality-Novelty:** 2
**Q2-2 Correctness-Technical Quality:** 2
**Q2-5 Clarity Of Writing:** 3

**Q1 Summary And Contributions:**

The work proposes a new methodology for building Bayesian neural nets (BNNs) based on the previous evidential deep learning framework, conjugate priors, the Laplace approximation, and normalizing flows. The driving idea is to assume that the classification task solved by the NNs corresponds to a problem of categorial random variables. This automatically leads to the assumption of Dirichlet priors for the NN weights. From that perspective, using an additional neural net to parameterize hyperparameters, it is claimed that uncertainty can be computed in one single pass and that normalizing flows can relax the issues that Dirichlet priors often add to the models.

**Q2-3 Extent To Which Claims Are Supported By Evidence:**

2: Fair: the main claims are somewhat supported by evidence (but the experimental evaluation may be weak, or does not match entirely with the claims, important baselines may be missing, proofs contain important ideas but lack rigor, algorithmic details are only discussed superficially, references are imprecise, assumptions are not sufficiently motivated or explicated, etc.).

**Q2-4 Reproducibility:**

2: Fair: key resources (e.g. proofs, code, data) are unavailable but key details (e.g. proof sketches, experimental setup) are sufficiently well-described for an expert to confidently reproduce the main results.

**Q3 Main Strengths:**

The submission is somehow clear and easy to follow. I particularly find well-written the review and use of the evidential deep learning [Ulmer, 2021] framework. In this direction, I also see strengths in the use of conjugate priors and the way that authors propose to advance in the context of BNNs. I would like to remark that the work contains interesting ideas, despite I find some points of weakness that I will point out later.

Proposition 4.1.3. looks great to me, even if not a lot of analysis is included about how the uncertainty error bounds increase or decrease wrt the number of classes C and the divergence d.

The amount of experiments included and the SOTA methods included to compare the proposed method seems to be in a good path, however, I don't really obtain a clear idea of the "good" performance of the method from them.

**Q4 Main Weakness:**

A short list of weaknesses or points with a 'lack of clarity' (in my opinion) that makes me doubt or be skeptical about the viability of the method for solving the main issues of BNNs:

**1)** The assumption of the Dirac delta to transform the parameter $\lambda(x, \psi)$ into the probability density function for integrating the posterior is somehow odd to me, and I'm not 100% sure if this really complies with Bayesian principles, even if it refers to the likelihood model.

**2)** The categorical distribution is announced in many ways throughout the paper, however, little is commented about the main weakness of the Categorical-Dirichlet conjugate system (its inefficacy when the number of classes C augment). If we assume, for instance, a Gumbel-Softmax model, how this can be integrated within the Dirichlet priors?

**3)** I'm not sure if the assumption in Eq. (6), where the hyperparameters of the Dirichlet prior are produced by an additional NN($\theta$) makes the problem simpler or more complicated. For sure, it is a bit overparametrized and little discussion about the size of this net is added.

**4)** Given the hyper-hyper-parameters $\theta$ (they are used to generate hyperparameters $\alpha$ of the Dirichlet prior), I'm somehow surprised about the Laplace approximation being fitted on top of the hyperparameters, or which parameters are we tuning here to find the MAP point-estimate? This is certainly not similar to previous LA approaches to BNNs.

**5)** Proposition 4.1.3 looks right, but I'm somehow concerned about how it increases wrt to the number of classes C and the divergence. The KL usually 'explodes' very quickly to large values, while the number of classes can also get large to 3 orders of magnitude, for instance. Could some analysis be included?

**Q5 Detailed Comments To The Authors:**

I added some questions and concerns together with the weaknesses. It would be really useful to get them answered somehow.

**Q9 Complying With Reviewing Instructions:**

Yes

---

> ### Author Rebuttal · Authors · 2024-04-07
>
> We appreciate all the detailed comments. Please find our responses below.
>
> **Output-space Posterior**: For a standard classification model, the softmax probability $\lambda(x,\psi)$ for input $x$, is a fixed vector for a given $\psi$. Therefore, $p(\lambda|x,\psi)$ follows a Dirac distribution, which is inherent to the BNN's architecture rather than an assumption. Eq. 2 is mathematically correct and writing in this way can facilitate comparison with Dirichlet-based networks.
>
> **Categorical Distribution**: The Dirichlet distribution is the standard conjugate prior for the categorical distribution. We have discussed its weaknesses in Sec. 4.3 such as complicating optimization, undefined log-likelihood for zero observations, and biased maximum likelihood estimation.  We will further emphasize these limitations, particularly when $C$ is large. Additionally, we propose to generalize the Dirichlet distribution using expressive NF to overcome the limitations. Notably, Dirichlet-based UQ methods are scalable to at least hundreds of classes, as evidenced by a summary of datasets used in this field in Appendix B’s Table 5 of [Ulmer, 2021].
>
> **Gumbel-softmax model**: This model primarily approximates the categorical distribution. It employs a reparameterization trick to approximately generate samples from the categorical distribution. It becomes unnecessary when sampling from the categorical distribution of the target $y$ is not required for both training and UQ. Despite this, we can further integrate the Gumbel-Softmax with a Dirichlet prior in Eq. 6. Eq. 6 can approximate $p(y|\alpha(x,\theta))$ using the Gumbel-Softmax model with parameters $(\frac{\alpha_i}{\alpha_0})_{i=1}^C$. Here, a sample $y^s = [y_1^s, \ldots, y_C^s]^T \sim p(y|\alpha(x,\theta))$ is calculated as follows:
>
> $$y_i^s = \frac{\exp((\log(\frac{\alpha_i}{\alpha_0}) +g_i^s )/\tau)}{\sum_{j=1}^C \exp((\log(\frac{\alpha_j}{\alpha_0}) +g_j^s )/\tau) } ~ i=1,…C, g_i^s, g_j^s \sim Gumble(0,1)$$
>
> where $\tau$ is a small constant.
>
> **Net Size**: In our hierarchical model $\theta \rightarrow \alpha \rightarrow \lambda \rightarrow y$, only $\theta$ is learnable, with other parameters explicitly calculated. Unlike standard NN training ($\theta \rightarrow \alpha \rightarrow y$, where alpha denotes NN output), adding Dirichlet $\lambda$ doesn't complicate the process due to closed-form integration. The NN size remains the same to a standard deterministic NN, followed by our experiments. To clarify, training directly with the likelihood in Eq. 6 resembles standard training, aiming to align $p(y|\alpha(x,\theta))$ with ground truth labels. However, it can only solve $\alpha$ up to a scale factor $\alpha_0$. Determining $\alpha_0$ (which is important for epistemic UQ) does not need to complicate the NN. We will include further discussions.
>
> **LA**: LA constructs a Gaussian distribution for the NN weights $\theta$ given a likelihood $p(y|x,\theta)$ (for computing $p(\mathcal{D}|\theta)$) and a prior $p(\theta)$. In our case, we apply LA on $\theta$ given the likelihood $p(y|x,\theta) = p(y|\alpha(x,\theta))$ in Eq. 6 and a prior $p(\theta)$, aligning closely with standard LA. The tuning parameter is $\theta$ which is the only learnable parameter in our model.
>
> **Proposition 4.1.3**: We treat $C$ as a constant and will analyze the bounds solely in terms of changes in $d$. $d$ often does not "explode" as it is defined on the probability simplex in the output space, which is much smaller than the parameter space. $d$ can remain small if $p(\lambda|x,\theta)$ is effectively trained to approximate $p(\lambda|x,\mathcal{D})$. Two analyses will be presented: (1) under the assumptions illustrated in Eq. 53 of Appendix B.4.3, all bounds decrease as $d$ decreases; (2) all bounds approach 0 as $d \rightarrow 0$. It is evident that the bound for $U_a$ satisfies the above conditions. Since the bound for $U_e$ is the sum of the bounds for $U_t$ and $U_a$, we only need to demonstrate that the bound for $U_t$ meets these conditions. Specifically, based on Lemma B.2, the function $-x\log \frac{x}{C}$ (with $x=\sqrt{2\log 2d}$ for the bound) is monotonically non-decreasing for $x \in [0,\frac{C}{e}]$. As $d$ approaches 0, the bound tends to 0, according to L'Hôpital's Rule:
>
> $$
> \lim_{{x\rightarrow 0}} -x\log \frac{x}{C} = \lim_{{x\rightarrow 0}} \frac{\log \frac{C}{x}}{\frac{1}{x}} = \lim_{{x\rightarrow 0}} \frac{\frac{d}{dx}\log \frac{C}{x}}{\frac{d}{dx}\frac{1}{x}} = \lim_{{x\rightarrow 0}} \frac{-1/x}{-1/x^2} = \lim_{{x\rightarrow 0}} x = 0
> $$
>
> **“Good” performance**: Compared to other Dirichlet-based methods such as PriorNet, PostNet, and EviNet, our method demonstrates a more significant improvement. When compared with single-network UQ methods like DUQ and DDU, our method also shows noticeable improvement. In comparison to ensemble methods, we achieve only comparable results. However, our method leverages single, sampling-free evidential networks that can enable fast UQ.

---

### Official Review · Reviewer_2A8Z · 2024-03-28

**Q2-1 Originality-Novelty:** 4
**Q2-2 Correctness-Technical Quality:** 2
**Q2-5 Clarity Of Writing:** 3

**Q1 Summary And Contributions:**

This paper proposes a Dirichlet-based method to approximate the Bayesian predictive distribution $p(\lambda|x,D)$ in the output space. Specifically, a Dirichlet distribution in the output space $p(\lambda|\alpha(x,\theta))$ is optimized to fit $p(\lambda|x,D)$, with the supervision from BNNs (ensemble models). The empirical results on OOD detection and calibration show improved uncertainty estimation.

**Q2-3 Extent To Which Claims Are Supported By Evidence:**

3: Good: the main claims are supported by convincing evidence (in the form of adequate experimental evaluation, proofs, (pseudo-)code, references, assumptions).

**Q2-4 Reproducibility:**

3: Good: key resources (e.g. proofs, code, data) are available and key details (e.g. proofs, experimental setup) are sufficiently well-described for competent researchers to confidently reproduce the main results.

**Q3 Main Strengths:**

1. A nice idea to directly approximate the $p(\lambda|x,D)$ rather than computing the BMA integral.

2. Sufficient theoretical guarantees on the effectiveness of uncertainty estimation algorithm.

3. Outstanding empirical results.

**Q4 Main Weakness:**

1. Still need an external BNN model as supervision to obtain good performances.

2. Missing some important experiments.

3. Some theoretical results are hard to interpret.

4. Missing some related works.

**Q5 Detailed Comments To The Authors:**

1. The proposed method needs an external BNN as supervision. I'm not sure if this setting is original. It seems too costly to train another model when you already have a converged BNN.

2. This paper is missing MCMC baselines like [1, 2, 3], which are considered the most gold-standard algorithm for Bayesian inference. This paper only considers comparing to ensembles and LA, which is not sufficient to demonstrate its effectiveness.

3. The bounds in Proposition 4.1.3 are confusing. It is hard to conclude whether the differences would be smaller when the KL-divergence becomes smaller.

4. For the ensemble-based UQ methods, some important related works are missing, like [4, 5].

[1] https://icml.cc/2011/papers/398_icmlpaper.pdf

[2] https://proceedings.mlr.press/v32/cheni14.pdf

[3] https://openreview.net/pdf?id=oGNdBvymod

[4] https://openreview.net/pdf?id=OOsR8BzCnl5

[5] https://openaccess.thecvf.com/content/CVPR2022/papers/Li_Trustworthy_Long-Tailed_Classification_CVPR_2022_paper.pdf

**Q9 Complying With Reviewing Instructions:**

Yes

---

> ### Author Rebuttal · Authors · 2024-04-07
>
> Thank you so much for the detailed review. We appreciate all the comments. Please find our responses below.
>
> **Requirements of BNN**: While the use of Bayesian models for knowledge distillation is not new, as discussed in Sec. 2, to our knowledge, we are the first to propose output-space distillation. This involves distilling Bayesian models into Dirichlet-based networks, accompanied by a theoretical analysis. The primary goal of the Dirichlet-based method is to enhance the efficiency of UQ. While a BNN is needed for training, single-network sampling-free Dir-BNN enables real-time inference where uncertainty can be captured within a single forward pass, making it suitable for many real-world applications due to its fast and accurate UQ. The BNN-augmented Dir-BNN can further perform UQ with theoretical guarantees. Without BNNs, previous evidential UQ methods fall short in demonstrating their capture of epistemic uncertainty. While getting an exact BNN may be costly, we propose the Dir-LA approach for efficient network training via self-distillation.
>
> **MCMC Baselines**: MCMC baselines have not been considered due to their inefficiency in both training and testing. Since the primary goal of Dir-BNN is to achieve fast inference while maintaining high accuracy, we have primarily compared our methods with other single-network UQ methods that can estimate uncertainty within a single forward pass. However, we acknowledge the reviewer's suggestion that adding MCMC baselines could better demonstrate the effectiveness of our proposed method. Consequently, we have also compared our method with SGLD [1] and SGHMC [2] for OOD detection on the MNIST and FMNIST datasets. We followed the experimental settings outlined in Appendix E.1 to ensure consistency in model architecture and some training protocols. For the burn-in period, we used 50 epochs. After the burn-in period, we selected 50 samples for each method every two epochs. The results are presented below (AUROC/AUPR), where our method continues to perform competitively. More analysis will be added in the revised paper.
>
> Method | MNIST -> Omniglot | MNIST -> KMNIST | FMNIST -> EMNIST | FMNIST -> MNIST |
> --- | --- | --- | --- | --- |
> Dir-LA | 97.9/97.7 | 98.7/98.5 | 91.8/95.2 | 94.4/93.9 |
> SGLD | 98.0/97.7 | 99.5/99.5 | 87.1/88.8 | 90.5/91.7 |
> SGHMC | 98.2/98.1 | 99.3/99.3 | 89.7/91.2 | 95.1/95.7 |
>
> **Bounds in Proposition 4.1.3**:  We will further present two analyses for the bounds: (1) under the assumptions illustrated in Eq. (53) of Appendix B.4.3, all bounds decrease as $d$ decreases; (2) all bounds approach 0 as $d \rightarrow 0$. It is evident that the bound for $U_a$ satisfies the above conditions. Since the bound for $U_e$ is the sum of the bounds for $U_t$ and $U_a$ (i.e., we use $|U_e - U_e^*| \leq |U_t - U_t^*| + |U_a - U_a^*|$ for Proposition 4.1.3.), we only need to demonstrate that the bound for $U_t$ meets these conditions. Based on Lemma B.2, the function $-x\log \frac{x}{C}$ (with $x=\sqrt{2\log 2d}$ for the bound of $U_t$) is monotonically non-decreasing if $x \in [0,\frac{C}{e}]$. Therefore, under the assumptions illustrated in Eq. (53) of Appendix B.4.3, this bound of $U_t$ is non-decreasing. As $d$ approaches 0, the bound tends to 0, according to L'Hôpital's Rule. For example,
>
> $$\lim_{x\rightarrow 0} -x\log \frac{x}{C} = \lim_{x\rightarrow 0} \frac{\log \frac{C}{x}}{\frac{1}{x}} = \lim_{x\rightarrow 0} \frac{\frac{d}{dx}\log \frac{C}{x}}{\frac{d}{dx}\frac{1}{x}} = \lim_{x\rightarrow 0} \frac{-1/x }{-1/x^2} =\lim_{x\rightarrow 0} x = 0$$
>
> It's also worth noting that the assumptions in Eq. (53) provide an upper bound for $d$, which may generally hold as $p(\lambda|x,\theta)$ is trained effectively to approximate $p(\lambda|x,\mathcal{D})$. $d$ often does not “explode” since the distributions are defined on the probability simplex in the output space, which is much smaller than the parameter space.
>
> **Ensemble-based Reference**: we will cite those papers and discuss them in the revised paper.

---

### Meta-Review · Area_Chair_TDcS · 2024-04-16

The paper proposes a Dirichlet-based model that integrates the advantages of Bayesian methods and evidential learning. The proposed method distills knowledge from Bayesian neural networks into an evidential model, enabling uncertainty estimation in a single forward pass. To enhance the distillation performance, the model further applies Laplacian approximation and normalizing flow techniques. The main concerns from the reviewers are about the clarity of some assumptions and claims, and the method's compatibility with a broader range of Bayesian neural networks. I encourage the authors to revise the paper according to the reviews.